# STOCHASTIC TRAINING OF GRAPH CONVOLUTIONAL NETWORKS

## ABSTRACT

Graph convolutional networks (GCNs) are powerful deep neural networks for graph-structured data. However, GCN computes nodes' representation recursively from their neighbors, making the receptive field size grow exponentially with the number of layers. Previous attempts on reducing the receptive field size by sub-sampling neighbors do not have any convergence guarantee, and their receptive field size per node is still in the order of hundreds. In this paper, we develop a preprocessing strategy and two control variate based algorithms to further reduce the receptive field size. Our algorithms are guaranteed to converge to GCN's local optimum regardless of the neighbor sampling size. Empirical results show that our algorithms have a similar convergence speed per epoch with the exact algorithm even using only two neighbors per node. The time consumption of our algorithm on the Reddit dataset is only one fifth of previous neighbor sampling algorithms.

## 1 INTRODUCTION

Graph convolution networks (GCNs) (Kipf & Welling, 2017) generalize convolutional neural networks (CNNs) (LeCun et al., 1995) to graph structured data. The "graph convolution" operation applies same linear transformation to all the neighbors of a node, followed by mean pooling. By stacking multiple graph convolution layers, GCNs can learn nodes' representation by utilizing information from distant neighbors. GCNs have been applied to semi-supervised node classification (Kipf & Welling, 2017), inductive node embedding (Hamilton et al., 2017a), link prediction (Kipf & Welling, 2016; Berg et al., 2017) and knowledge graphs (Schlichtkrull et al., 2017), outperforming multi-layer perceptron (MLP) models that do not use the graph structure and graph embedding approaches (Perozzi et al., 2014; Tang et al., 2015; Grover & Leskovec, 2016) that do not use node features.

However, the graph convolution operation makes it difficult to train GCN efficiently. A node's representation at layer $L$ is computed recursively by all its neighbors' representations at layer $L-1$. Therefore, the receptive field of a single node grows exponentially with respect to the number of layers, as illustrated in Fig. 1(a). Due to the large receptive field size, Kipf & Welling (2017) proposed training GCN by a batch algorithm, which computes the representation for all the nodes altogether. However, batch algorithms cannot handle large scale datasets because of their slow convergence and the requirement to fit the entire dataset in GPU memory.

Hamilton et al. (2017a) made an initial attempt on developing stochastic algorithms to train GCNs, which is referred as neighbor sampling (NS) in this paper. Instead of considering all the neighbors, they randomly subsample $D^{(l)}$ neighbors at the $l$-th layer. Therefore, they reduce the receptive field size to $\prod_l D^{(l)}$, as shown in Fig. 1(b). They found that for two layer GCNs, keeping $D^{(1)} = 10$ and $D^{(2)} = 25$ neighbors can achieve comparable performance with the original model. However, there is no theoretical guarantee on the predictive performance of the model learnt by NS comparing with the original algorithm. Moreover, the time complexity of NS is still $D^{(1)}D^{(2)} = 250$ times larger than training an MLP, which is unsatisfactory.

In this paper, we develop novel stochastic training algorithms for GCNs such that $D^{(l)}$ can be as low as two, so that the time complexity of training GCN is comparable with training MLPs. Our methods are built on two techniques. First, we propose a strategy which preprocesses the first graph convolution layer, so that we only need to consider all neighbors within $L-1$ hops instead of $L$ hops. This is significant because most GCNs only have $L = 2$ layers (Kipf & Welling, 2017; Hamilton

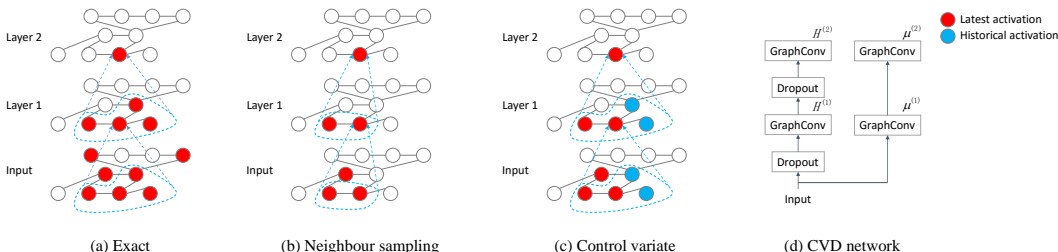

Figure 1: Two-layer graph convolutional networks, and the receptive field of a single vertex.

et al., 2017a). Second, we develop two control variate (CV) based stochastic training algorithms. We show that our CV-based algorithms have lower variance than NS, and for GCNs without dropout, our algorithm provably converges to a local optimum of the model regardless of $D^{(l)}$.

We empirically test on six graph datasets, and show that our techniques significantly reduce the bias and variance of the gradient from NS with the same receptive field size. Our algorithm with $D^{(l)} = 2$ achieves the same predictive performance with the exact algorithm in comparable number of epochs on all the datasets, while the training time is 5 times shorter on our largest dataset.

## 2 BACKGROUNDS

We now briefly review graph convolutional networks (GCNs) (Kipf & Welling, 2017) and the neighbor sampling (NS) algorithm (Hamilton et al., 2017a).

### 2.1 GRAPH CONVOLUTIONAL NETWORKS

The original GCN was presented in a semi-supervised node classification task (Kipf & Welling, 2017). We follow this setting throughout this paper. Generalization of GCN to other tasks can be found in Kipf & Welling (2016); Berg et al. (2017); Schlichtkrull et al. (2017) and Hamilton et al. (2017b). In the node classification task, we have an undirected graph $\mathcal{G} = (\mathcal{V}, \mathcal{E})$ with $V = |\mathcal{V}|$ vertices and $E = |\mathcal{E}|$ edges, where each vertex $v$ consists of a feature vector $x_v$ and a label $y_v$. The label is only observed for some vertices $\mathcal{V}_L$ and we want to predict the label for the rest vertices $\mathcal{V}_U := \mathcal{V} \backslash \mathcal{V}_L$. The edges are represented as a symmetric $V \times V$ adjacency matrix $A$, where $A_{v,v'}$ is the weight of the edge between $v$ and $v'$, and the propagation matrix $P$ is a normalized version of $A$: $\tilde{A} = A + I$, $\tilde{D}_{vv} = \sum_{v'} \tilde{A}_{vv'}$, and $P = \tilde{D}^{-\frac{1}{2}} \tilde{A} \tilde{D}^{-\frac{1}{2}}$. A graph convolution layer is defined as

$$\tilde{H}^{(l)} = \text{Dropout}_p(H^{(l)}), \quad Z^{(l+1)} = P\tilde{H}^{(l)}W^{(l)}, \quad H^{(l+1)} = \sigma(Z^{l+1}), \quad (1)$$

where $H^{(l)}$ is the activation matrix in the $l$-th layer, whose each row is the activation of a graph node. $H^{(0)} = X$ is the input feature matrix, $W^{(l)}$ is a trainable weight matrix, $\sigma(\cdot)$ is an activation function, and $\text{Dropout}_p(\cdot)$ is the dropout operation (Srivastava et al., 2014) with keep probability $p$. Finally, the loss is defined as $\mathcal{L} = \frac{1}{|\mathcal{V}_L|} \sum_{v \in \mathcal{V}_L} f(y_v, Z_v^{(L)})$, where $f(\cdot, \cdot)$ can be the square loss, cross entropy loss, etc., depending on the type of the label.

When $P = I$, GCN reduces to a multi-layer perceptron (MLP) model which does not use the graph structure. Comparing with MLP, GCN is able to utilize neighbor information for node classification. We define $\mathbf{n}(v, L)$ as the set of all the $L$-neighbors of node $v$, i.e., the nodes that are reachable from $v$ within $L$ hops. It is easy to see from Fig. 1(a) that in an $L$-layer GCN, a node uses the information from all its $L$-neighbors. This makes GCN more powerful than MLP, but also complicates the stochastic training, which utilizes an approximated gradient $\nabla \mathcal{L} \approx \frac{1}{|\mathcal{V}_B|} \sum_{v \in \mathcal{V}_B} \nabla f(y_v, Z_v^{(L)})$, where $\mathcal{V}_B \subset \mathcal{V}_L$ is a minibatch of training data. The large receptive field size $|\cup_{v \in \mathcal{V}_B} \mathbf{n}(v, L)|$ per minibatch leads to high time complexity, space complexity and amount of IO. See Table 1 for the average number of 1- and 2-neighbors of our datasets.

### 2.2 ALTERNATIVE NOTATION

We introduce alternative notations to help compare different algorithms. Let $U^{(l)} = P\tilde{H}^{(l)}$, or $u_v^{(l)} = \sum_{v' \in \mathbf{n}(v,1)} P_{v,v'} \tilde{h}_{v'}^{(l)}$, we focus on studying how $u_v$ is computed based on node $v$'s neighbors. To keep notations simple, we omit all the subscripts and tildes, and exchange the ID of nodes such

| Dataset | $V$ | $E$ | Degree | Degree 2 | Type |
|---------|-----|-----|--------|----------|------|
| Citeseer | 3,327 | 12,431 | 4 | 15 | Document network |
| Cora | 2,708 | 13,264 | 5 | 37 | Document network |
| PubMed | 19,717 | 108,365 | 6 | 60 | Document network |
| NELL | 65,755 | 318,135 | 5 | 1,597 | Knowledge graph |
| PPI | 14,755 | 458,973 | 31 | 970 | Protein-protein interaction |
| Reddit | 232,965 | 23,446,803 | 101 | 10,858 | Document network |

Table 1: Number of vertexes, edges, average number of 1- and 2-neighbors per node for each dataset. Undirected edges are counted twice and self-loops are counted once. Reddit is already subsampled to have a max degree of 128 following Hamilton et al. (2017a).

that $\mathbf{n}(v, 1) = [D]_+$, [1] where $D = |\mathbf{n}(v, 1)|$ is the number of neighbors. We get the propagation rule $u = \sum_{v=1}^{D} p_v h_v$, which is used interchangeably with the matrix form $U^{(l)} = P\tilde{H}^{(l)}$.

## 2.3 NEIGHBOR SAMPLING

To reduce the receptive field size, Hamilton et al. (2017a) propose a neighbor sampling (NS) algorithm. On the $l$-th layer, they randomly choose $D^{(l)}$ neighbors for each node, and develop an estimator $u_{NS}$ of $u$ based on Monte-Carlo approximation $u \approx u_{NS} = \frac{D}{D^{(l)}} \sum_{v \in \mathbf{D}^{(l)}} p_v h_v$, where $\mathbf{D}^{(l)} \subset [D]_+$ is a subset of $D^{(l)}$ neighbors. In this way, they reduce the receptive field size from $|\cup_{v \in \mathcal{V}_B} \mathbf{n}(v, L)|$ to $O(|\mathcal{V}_B| \prod_{l=1}^{L} D^{(l)})$. Neighbor sampling can also be written in a matrix form as

$$\tilde{H}_{NS}^{(l)} = \text{Dropout}_p(H_{NS}^{(l)}), \quad Z_{NS}^{(l+1)} = \hat{P}^{(l)} \tilde{H}_{NS}^{(l)} W^{(l)}, \quad H_{NS}^{(l+1)} = \sigma(Z_{NS}^{(l+1)}), \quad (2)$$

where $\hat{P}^{(l)}$ is a sparser unbiased estimator of $P$, i.e., $\mathbb{E}\hat{P}^{(l)} = P$. The approximate prediction $Z_{NS}^{(L)}$ used for testing and for computing stochastic gradient $\frac{1}{|\mathcal{V}_B|} \sum_{v \in \mathcal{V}_B} \nabla f(y_v, Z_{CV,v}^{(L)})$ during training.

The NS estimator $u_{NS}$ is unbiased. However it has a large variance, which leads to biased prediction and gradients after the non-linearity in subsequent layers. Due to the biased gradients, training with NS does not converge to the local optimum of GCN. When $D^{(l)}$ is moderate, NS may has some regularization effect like dropout (Srivastava et al., 2014), where it drops neighbors instead of features. However, for the extreme ease $D^{(l)} = 2$, the neighbor dropout rate is too high to reach high predictive performance, as we will see in Sec. 5.4. Intuitively, making prediction solely depends on one neighbor is inferior to using all the neighbors. To keep comparable prediction performance with the original GCN, Hamilton et al. (2017a) use relatively large $D^{(1)} = 10$ and $D^{(2)} = 25$. Their receptive field size $D^{(1)} \times D^{(2)} = 250$ is still much larger than MLP, which is 1.

## 3 PREPROCESSING FIRST LAYER

We first present a technique to preprocess the first graph convolution layer, by approximating $A\text{Dropout}_p(X)$ with $\text{Dropout}_p(AX)$. The model becomes

$$Z^{(l+1)} = \text{Dropout}_p(PH^{(l)})W^{(l)}, \quad H^{(l+1)} = \sigma(Z^{l+1}). \quad (3)$$

This approximation does not change the expectation because $\mathbb{E}\left[A\text{Dropout}_p(X)\right] = \mathbb{E}\left[\text{Dropout}_p(AX)\right]$, and it does not affect the predictive performance, as we shall see in Sec. 5.1.

The advantage of this modification is that we can preprocess $U^{(0)} = PH^{(0)} = PX$ and takes $U^{(0)}$ as the new input. In this way, the actual number of graph convolution layers is reduced by one — the first layer is merely a fully connected layer instead of a graph convolution one. Since most GCNs only have two graph convolution layers (Kipf & Welling, 2017; Hamilton et al., 2017a), this gives a significant reduction of the receptive field size from the number of $L$-neighbors $|\cup_{v \in \mathcal{V}_B} \mathbf{n}(v, L)|$ to the number of $L-1$-neighbors $|\cup_{v \in \mathcal{V}_B} \mathbf{n}(v, L-1)|$. The numbers are reported in Table 1.

## 4 CONTROL VARIATE BASED STOCHASTIC APPROXIMATION

We now present two novel control variate based estimators that have smaller variance as well as stronger theoretical guarantees than NS.

---

[1] For an integer $N$, we define $[N] = \{0, \ldots, N\}$ and $[N]_+ = \{1, \ldots, N\}$.

### 4.1 Control variate based estimator

We assume that the model does not have dropout for now and will address dropout in Sec. 4.2. The idea is that we can approximate $u = \sum_{v=1}^{D} p_v h_v$ better if we know the latest historical activations $\bar{h}_v$ of the neighbors, where we expect $\bar{h}_v$ and $h_v$ are similar if the model weights do not change too fast during the training. With the historical activations, we approximate

$$u = \sum_{v=1}^{D} p_v h_v = \sum_{v=1}^{D} p_v(h_v - \bar{h}_v) + \sum_{v=1}^{D} p_v \bar{h}_v \approx D p_{v'} \Delta h_{v'} + \sum_{v=1}^{D} p_v \bar{h}_v := u_{CV}, \qquad (4)$$

where $v'$ is a random neighbor, and $\Delta h_{v'} = h_{v'} - \bar{h}_{v'}$. For the ease of presentation, we assume that we only use the latest activation of one neighbor, while the implementation also include the node itself besides the random neighbor, so $D^{(l)} = 2$. Using historical activations is cheap because they need not to be computed recursively using their neighbors' activations, as shown in Fig. 1(c). Unlike NS, we apply Monte-Carlo approximation on $\sum_v p_v \Delta h_v$ instead of $\sum_v p_v h_v$. Since we expect $h_v$ and $\bar{h}_v$ to be close, $\Delta h_v$ will be small and $u_{CV}$ should have a smaller variance than $u_{NS}$. Particularly, if the model weight is kept fixed, $\bar{h}_v$ should be eventually equal with $h_v$, so that $u_{CV} = 0 + \sum_{v=1}^{D} p_v \bar{h}_v = \sum_{v=1}^{D} p_v h_v = u$, i.e., the estimator has zero variance. The term $CV = u_{CV} - u_{NS} = -D p_{v'} \bar{h}_{v'} + \sum_{v=1}^{D} p_v \bar{h}_v$ is a *control variate* (Ripley, 2009, Chapter 5), which has zero mean and large correlation with $u_{NS}$, to reduce its variance. We refer this stochastic approximation algorithm as CV, and we will formally analyze the variance and prove the convergence of the training algorithm using CV for stochastic gradient in subsequent sections.

In matrix form, CV computes the approximate predictions as follows, where we explicitly write down the iteration number $_i$ and add the subscript $_{CV}$ to the approximate activations [2]

$$Z_{CV,i}^{(l+1)} \leftarrow \left( \hat{P}_i^{(l)}(H_{CV,i}^{(l)} - \bar{H}_{CV,i}^{(l)}) + P\bar{H}_{CV,i}^{(l)} \right) W_i^{(l)}, \qquad (5)$$

$$H_{CV,i}^{(l+1)} \leftarrow \sigma(Z_{CV,i}^{(l+1)}), \quad \bar{H}_{CV,i+1}^{(l)} \leftarrow \mathbf{m}_i^{(l)} H_{CV,i}^{(l)} + (1 - \mathbf{m}_i^{(l)})\bar{H}_{CV,i}^{(l)}, \qquad (6)$$

where $\bar{h}_{CV,i,v}^{(l)}$ stores the latest activation of node $v$ on layer $l$ computed before time $i$. Formally, let $\mathbf{m}_i^{(l)} \in \mathbb{R}^{V \times V}$ be a diagonal matrix, and $(m_i^{(l)})_{vv} = 1$ if $(\hat{P}_i^{(l)})_{v'v} > 0$ for any $v'$. After finishing one iteration we update history $\bar{H}$ with the activations computed in that iteration as Eq. (6).

### 4.2 Control variate for dropout

With dropout, the activations $H$ are no longer deterministic. They become random variables whose randomness come from different dropout configurations. Therefore, $\Delta h_v = h_v - \bar{h}_v$ is not necessarily small even if $h_v$ and $\bar{h}_v$ have the same distribution. We develop another stochastic approximation algorithm, *control variate for dropout* (CVD), that works well with dropout.

Our method is based on the weight scaling procedure (Srivastava et al., 2014) to approximately compute the mean $\mu_v := \mathbb{E}[h_v]$. That is, along with the dropout model, we can run a copy of the model with no dropout to obtain the mean $\mu_v$, as illustrated in Fig. 1(d). With the mean, we can obtain a better stochastic approximation by separating the mean and variance

$$u = \sum_{v=1}^{D} p_v \left[ (h_v - \mu_v) + (\mu_v - \bar{\mu}_v) + \bar{\mu}_v \right] \approx \sqrt{D} p_{v'}(h_{v'} - \mu_{v'}) + D p_{v'} \Delta \mu_{v'} + \sum_{v=1}^{D} p_v \bar{\mu}_v := u_{CVD},$$

where $\bar{\mu}_v$ is the historical mean activation, obtained by storing $\mu_v$ instead of $h_v$, and $\Delta \mu = \mu_v - \bar{\mu}_v$. $u_{CVD}$ an unbiased estimator of $u$ because the term $\sqrt{D} p_{v'}(h_{v'} - \mu_{v'})$ has zero mean, and the Monte-Carlo approximation $\sum_{v=1}^{D} p_v(\mu_v - \bar{\mu}_v) \approx D p_{v'} \Delta \mu_{v'}$ does not change the mean. The approximation $\sum_{v=1}^{D} p_v(h_v - \mu_v) \approx \sqrt{D} p_{v'}(h_{v'} - \mu_{v'})$ is made by assuming $h_v$'s to be independent Gaussians, which we will soon clarify. The pseudocodes for CV and CVD are in Appendix E.

### 4.3 Variance analysis

NS, CV and CVD are all unbiased estimators of $u = \sum_v p_v h_v$. We analyze their variance in a simple independent Gaussian case, where we assume that activations are Gaussian random variables

---

[2] We will omit the subscripts $_{CV}$ and $_i$ in subsequent sections when there is no confusion.

| Alg. | Estimator | Var. from MC. approx. | Var. from dropout |
|------|-----------|----------------------|-------------------|
| Exact | $u = \sum_v p_v h_v$ | $0$ | $s^2$ |
| NS | $u_{NS} = D p_{v'} h_{v'}$ | $\frac{1}{2} \sum_{v,v'} (p_v \mu_v - p_{v'} \mu_{v'})^2$ | $Ds^2$ |
| CV | $u_{CV} = D p_{v'} \Delta h_{v'} + \sum_v p_v \bar{h}_v$ | $\frac{1}{2} \sum_{v,v'} (p_v \Delta\mu_v - p_{v'} \Delta\mu_{v'})^2$ | $Ds^2 + (D-1)\bar{s}^2$ |
| CVD | $u_{CVD} = \sqrt{D} p_{v'} (h_{v'} - \mu_{v'})$ $+ D p_{v'} \Delta\mu_{v'} + \sum_v p_v \bar{\mu}_v$ | $\frac{1}{2} \sum_{v,v'} (p_v \Delta\mu_v - p_{v'} \Delta\mu_{v'})^2$ | $s^2$ |

Table 2: Variance of different algorithms in the independent Gaussian case.

$h_v \sim \mathcal{N}(\mu_v, s_v^2)$ following Wang & Manning (2013). Without loss of generality, we assume that all the activations $h_v$ are one dimensional. We also assume that all the activations $h_1, \ldots, h_D$ and historical activations $\bar{h}_1, \ldots, \bar{h}_D$ are independent, where the historical activations $\bar{h}_v \sim \mathcal{N}(\bar{\mu}_v, \bar{s}_v^2)$.

We introduce a few more notations. $\Delta\mu_v$ and $\Delta s_v^2$ are the mean and variance of $\Delta h_v = h_v - \bar{h}_v$, where $\Delta\mu_v = \mu_v - \bar{\mu}_v$ and $\Delta s_v^2 = s_v^2 + \bar{s}_v^2$. $\mu$ and $s^2$ are the mean and variance of $\sum_v p_v h_v$, where $\mu = \sum_v p_v \mu_v$ and $s^2 = \sum_v p_v^2 s_v^2$. Similarly, $\Delta\mu$, $\Delta s^2$, $\bar{\mu}$ and $\bar{s}^2$ are the mean and variance of $\sum_v p_v \Delta h_v$ and $\sum_v p_v \bar{h}_v$, respectively.

With these assumptions and notations, we list the estimators and variances in Table 2, where the derivations can be found in Appendix C. We decompose the variance as two terms: variance from Monte-Carlo approximation (VMCA) and variance from dropout (VD).

If the model has no dropout, the activations have zero variance, i.e., $s_v = \bar{s}_v = 0$, and the only source of variance is VMCA. We want VMCA to be small. As in Table 2, the VMCA for the exact estimator is 0. For the NS estimator, VMCA is $\frac{1}{2} \sum_{v,v'} (p_v \mu_v - p_{v'} \mu_{v'})^2$, whose magnitude depends on the pairwise difference $(p_v \mu_v - p_{v'} \mu_{v'})^2$, and VMCA is zero if and only if $p_v \mu_v = p_{v'} \mu_{v'}$ for all $v, v'$. Similarly, VMCA for both CV and CVD estimators is $\frac{1}{2} \sum_{v,v'} (p_v \Delta\mu_v - p_{v'} \Delta\mu_{v'})^2$, which should be smaller than NS estimator's VMCA if $(p_v \Delta\mu_v - p_{v'} \Delta\mu_{v'})^2 < (p_v \mu_v - p_{v'} \mu_{v'})^2$, which is likely because $\Delta\mu_v$ should be smaller than $\mu_v$. Since CV and CVD estimators have the same VMCA, we adopt the CV estimator for models without dropout, due to its simplicity.

The VD of the exact estimator is $s^2$, which is overestimated by both NS and CV. NS overestimates VD by $D$ times, and CV has even larger VD. Meanwhile, the VD of the CVD estimator is the same as the exact estimator, indicating CVD to be the best estimator for models with dropout.

### 4.4 EXACT TESTING

Besides smaller variance, CV also has stronger theoretical guarantees than NS. We can show that during testing, CV's prediction becomes exact after a few testing epochs. For models without dropout, we can further show that training using the stochastic gradients obtained by CV converges to GCN's local optimum. We present these results in this section and Sec. 4.5. Note that the analysis does *not* need the independent Gaussian assumption.

Given a model $W$, we compare the exact predictions (Eq. 1) and CV's approximate predictions (Eq. 5,6) during testing, which uses the deterministic weight scaling procedure. To make predictions, we run forward propagation by epochs. In each epoch, we randomly partition the vertex set $\mathcal{V}$ as $I$ minibatches $\mathcal{V}_1, \ldots, \mathcal{V}_I$ and in the $i$-th iteration, we run a forward pass to compute the prediction for nodes in $\mathcal{V}_i$. Note that in each epoch we scan *all* the nodes instead of just *testing* nodes, to ensure that the activation of each node is computed at least once per epoch. The following theorem reveals the connection of the exact predictions and gradients, and their approximate versions by CV.

**Theorem 1.** *For a fixed $W$ and any $i > LI$ we have: (1) (Exact Prediction) The activations computed by CV are exact, i.e., $Z_{CV,i}^{(l)} = Z^{(l)}$ for each $l \in [L]$ and $H_{CV,i}^{(l)} = H^{(l)}$ for each $l \in [L-1]$. (2) (Unbiased Gradient) The stochastic gradient $g_{CV,i}(W) := \frac{1}{|\mathcal{V}_B|} \sum_{v \in \mathcal{V}_B} \nabla_W f(y_v, z_{CV,i,v}^{(L)})$ is an unbiased estimator of GCN's gradient, i.e., $\mathbb{E}_{\hat{P}, \mathcal{V}_B} g_{CV,i}(W) = \nabla_W \frac{1}{|V|} \sum_{v \in \mathcal{V}} f(y_v, z_v^{(L)})$.*

Theorem 1 shows that at testing time, we can run forward propagation with CV for $L$ epochs and get the exact prediction. This outperforms NS, which cannot recover the exact prediction. Comparing with directly making exact predictions by a batch algorithm, CV is more scalable because it does not need to load the entire graph into memory. The proof can be found in Appendix A.

### 4.5 CONVERGENCE GUARANTEE

The following theorem shows that for a model without dropout, training using CV's approximated gradients converges to a local optimum of GCN, regardless of the neighbor sampling size $D^{(l)}$. Therefore, we can choose arbitrarily small $D^{(l)}$ without worrying about the convergence.

**Theorem 2.** *Assume that (1) all the activations are $\rho$-Lipschitz, (2) the gradient of the cost function $\nabla_z f(y, z)$ is $\rho$-Lipschitz and bounded, (3) $\|g_{CV}(W)\|_\infty$ and $\|g(W)\|_\infty = \|\nabla \mathcal{L}(W)\|_\infty$ are bounded by $G > 0$ for all $\hat{P}, \mathcal{V}_B$ and $W$. (4) The loss $\mathcal{L}(W)$ is $\rho$-smooth, i.e., $|\mathcal{L}(W_2) - \mathcal{L}(W_1) - \langle \nabla L(W_1), W_2 - W_1 \rangle| \le \frac{\rho}{2} \|W_2 - W_1\|^2 \, \forall W_1, W_2$, where $\langle A, B \rangle = tr(A^\top B)$ is the inner product of matrix $A$ and matrix $B$. We randomly run SGD for $R \le N$ iterations, where $P_R(R = i) = \frac{2\gamma_i - \rho\gamma_i^2}{\sum_{i=1}^N (2\gamma_i - \rho\gamma_i^2)}$. Then, for the updates $W_{i+1} = W_i - \gamma_i g_{CV}(W_i)$ and step sizes $\gamma_i = \min\{\frac{1}{\rho}, \frac{1}{\sqrt{N}}\}$, there exists constants $K_1$ and $K_2$ which are irrelevant with $N$, s.t. $\forall N > LI$,*

$$\mathbb{E}_{R \sim P_R} \mathbb{E}_{\hat{P}, \mathcal{V}_B} \|\nabla \mathcal{L}(W_R)\|^2 \le \frac{2\rho(\mathcal{L}(W_1) - \mathcal{L}(W^*)) + K_2}{N} + \frac{2(\mathcal{L}(W_1) - \mathcal{L}(W^*)) + K_1}{\sqrt{N}}.$$

The proof can be found in Appendix B. Particularly, $\lim_{N \to \infty} \mathbb{E}_{R \sim P_R} \mathbb{E}_{\hat{P}, \mathcal{V}_B} \|\nabla \mathcal{L}(W_R)\|^2 = 0$. Therefore, our algorithm converges to a local optimum.

### 4.6 TIME COMPLEXITY AND IMPLEMENTATION DETAILS

Finally we discuss the time complexity of different algorithms. We decompose the time complexity as *sparse time complexity* for sparse-dense matrix multiplication such as $P\tilde{H}^{(l)}$, and *dense time complexity* for dense-dense matrix multiplication such as $U^{(l)}W^{(l)}$. Assume that the node feature is $K$-dimensional and the first hidden layer is $A$-dimensional, the batch GCN has $O(EK)$ sparse and $O(VKA)$ dense time complexity per epoch. NS has $O(V \prod_{l=1}^L D^{(l)} K)$ sparse and $O(V \prod_{l=2}^L D^{(l)} KA)$ dense time complexity per epoch. The dense time complexity of CV is the same as NS. The sparse time complexity depends on the cost of computing the sum $\sum_v p_v \bar{\mu}_v$. There are $V \prod_{l=2}^L D^{(l)}$ such sums to compute on the first graph convolution layer, and overall cost is not larger than $O(VD \prod_{l=2}^L D^{(l)} K)$, if we subsample the graph such that the max degree is $D$, following Hamilton et al. (2017a). The sparse time complexity is $D/D^{(1)}$ times higher than NS.

Our implementation is similar as Kipf & Welling (2017). We store the node features in the main memory, without assuming that they fit in GPU memory as Hamilton et al. (2017a), which makes our implementation about 2 times slower than theirs. We keep the histories in GPU memory for efficiency since they are only $LH < K$ dimensional.

## 5 EXPERIMENTS

We examine the variance and convergence of our algorithms empirically on six datasets, including Citeseer, Cora, PubMed and NELL from Kipf & Welling (2017) and Reddit, PPI from Hamilton et al. (2017a), as summarized in Table 1. To measure the predictive performance, we report Micro-F1 for the multi-label PPI dataset, and accuracy for all the other multi-class datasets. We use the same model architectures with previous papers but slightly different hyperparameters (see Appendix D for the details). We repeat the convergence experiments 10 times on Citeseer, Cora, PubMed and NELL, and 5 times on Reddit and PPI. The experiments are done on a Titan X (Maxwell) GPU.

### 5.1 IMPACT OF PREPROCESSING

We first examine the approximation in Sec. 3 that switches the order of dropout and aggregating the neighbors. Let M0 be the original model (Eq. 1) and M1 be our approximated model (Eq. 3), we compare three settings: (1) M0, $D^{(l)} = \infty$ is the exact algorithm without any neighbor sampling. (2) M1+PP, $D^{(l)} = \infty$ changes the model from M0 to M1. Preprocessing does not affect the training for $D^{(l)} = \infty$. (3) M1+PP, $D^{(l)} = 20$ uses NS with a relatively large number of neighbors. In Table 3 we can see that all the three settings performs similarly, i.e., our approximation does not affect the predictive performance. Therefore, we use M1+PP, $D^{(l)} = 20$ as the exact baseline in following convergence experiments because it is the fastest among these three settings.

| Algorithm | Citeseer | Cora | PubMed | NELL | PPI | Reddit |
|---|---|---|---|---|---|---|
| Epochs | 200 | 200 | 200 | 200 | 500 | 10 |
| M0, $D^{(l)} = \infty$ | $70.8 \pm .1$ | $81.7 \pm .5$ | $79.0 \pm .4$ | - | $97.9 \pm .04$ | $96.2 \pm .04$ |
| M1+PP, $D^{(l)} = \infty$ | $70.9 \pm .2$ | $82.0 \pm .8$ | $78.7 \pm .3$ | $64.9 \pm 1.7$ | $97.8 \pm .05$ | $96.3 \pm .07$ |
| M1+PP, $D^{(l)} = 20$ | $70.9 \pm .2$ | $81.9 \pm .7$ | $78.9 \pm .5$ | $64.2 \pm 4.6$ | $97.6 \pm .09$ | $96.3 \pm .04$ |

Table 3: Testing accuracy of different algorithms and models after fixed number of epochs. Our implementation does not support M0, $D^{(l)} = \infty$ on NELL so the result is not reported

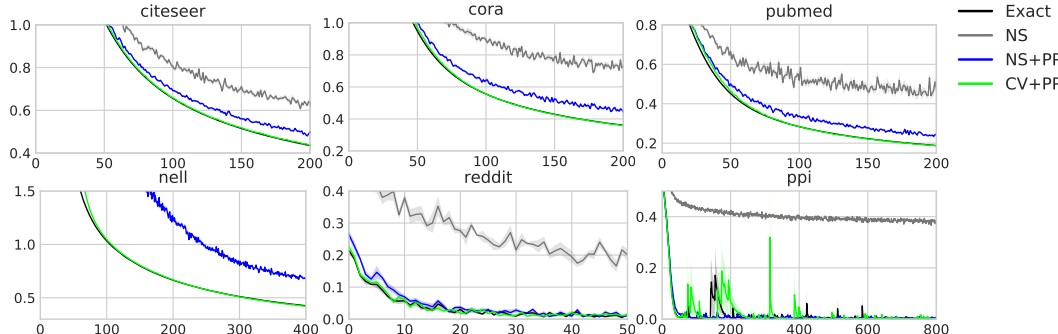

Figure 2: Comparison of training loss with respect to number of epochs without dropout. The CV+PP curve overlaps with the Exact curve in the first four datasets.

## 5.2 CONVERGENCE WITH NO DROPOUT

We now study how fast our algorithms converge with a very small neighbor sampling size $D^{(l)} = 2$. We compare the following algorithms: (1) Exact, which is M1+PP, $D^{(l)} = 20$ in Sec. 5.1 as a surrogate of the exact algorithm. (2) NS, which is the NS algorithm with no preprocessing and $D^{(l)} = 2$. (3) NS+PP, which is same with NS but uses preprocessing. (4) CV+PP, which replaces the NS estimator in NS+PP with the CV estimator. (5) CVD+PP, which uses the CVD estimator.

We first validate Theorem 2, which states that CV+PP converges to a local optimum of Exact, for models without dropout, regardless of $D^{(l)}$. We disable dropout and plot the training loss with respect to number of epochs as Fig. 2. We can see that CV+PP can always reach the same training loss with Exact, which matches the conclusion of Theorem 2. Meanwhile, NS and NS+PP have a higher training loss because their gradients are biased.

## 5.3 CONVERGENCE WITH DROPOUT

Next, we compare the predictive accuracy obtained by the model trained by different algorithms, with dropout turned on. We use different algorithms for training and the same Exact algorithm for testing, and report the validation accuracy at each training epoch. The result is shown in Fig. 3. We find that CVD+PP is the only algorithm that is able to reach comparable validation accuracy with Exact on all datasets. Furthermore, its convergence speed with respect to the number of epochs is comparable with Exact despite its $D^{(l)}$ is 10 times smaller. Note that CVD+PP performs much better than Exact on the PubMed dataset; we suspect it finds a better local optimum.

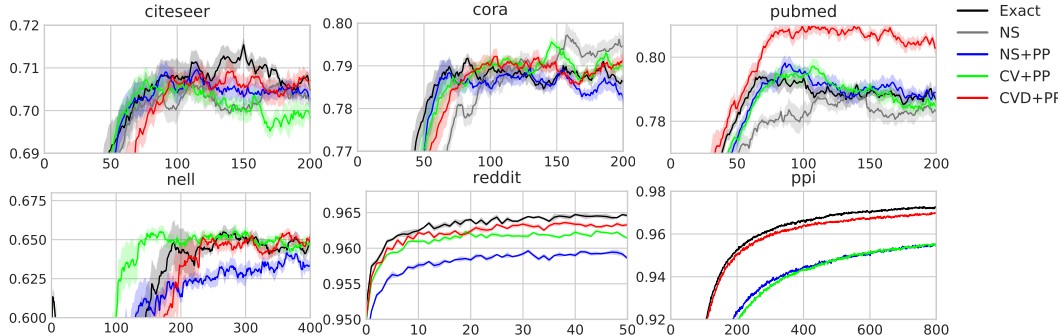

Figure 3: Comparison of validation accuracy with respect to number of epochs. NS converges to 0.94 on the Reddit dataset and 0.6 on the PPI dataset.

| Alg. | Valid. acc. | Epochs | Time (s) | Sparse GFLOP | Dense TFLOP |
|---|---|---|---|---|---|
| Exact | 96.0 | **4.2** | 252 | 507 | 7.17 |
| NS | 94.4 | 102.0 | 577 | 76.5 | 21.4 |
| NS+PP | 96.0 | 35.0 | 195 | **2.53** | 7.36 |
| CV+PP | 96.0 | 7.8 | 56 | 40.6 | **1.64** |
| CVD+PP | 96.0 | 5.8 | **50** | 60.3 | 2.44 |

Table 4: Time complexity comparison of different algorithms on the Reddit dataset.

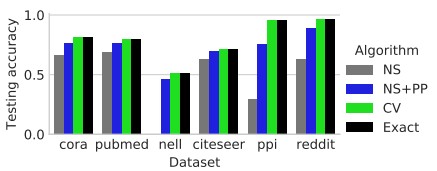

Figure 4: Comparison of the accuracy of different testing algorithms. The y-axis is Micro-F1 for PPI and accuracy otherwise.

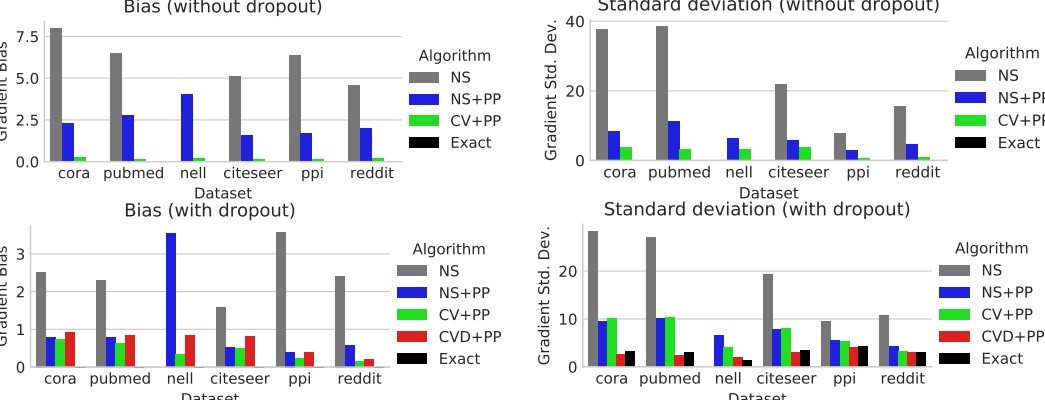

Figure 5: Bias and standard deviation of the gradient for different algorithms during training.

Meanwhile, simper algorithms CV+PP and NS+PP work acceptably on most of the datasets. CV+PP reaches a comparable accuracy with Exact for all datasets except PPI. NS+PP works slightly worse but the final validation accuracy is still within 2%. These algorithms can be adopted if there is no strong need for predictive performance. We however emphasize that exact algorithms must be used for making predictions, as we will show in Sec. 5.4. Finally, the algorithm NS without preprocessing works much worse than others, indicating the significance of our preprocessing strategy.

## 5.4 FURTHER ANALYSIS ON TIME COMPLEXITY, TESTING ACCURACY AND VARIANCE

Table 4 reports the average number of epochs, time, and total number of floating point operations to reach a given 96% validation accuracy on the largest Reddit dataset. Sparse and dense computations are defined in Sec. 4.6. We found that CVD+PP is about 5 times faster than Exact due to the significantly reduced receptive field size. Meanwhile, simply setting $D^{(l)} = 2$ for NS does not converge to the given accuracy.

We compare the quality of the predictions made by different algorithms, using the *same* model trained by Exact in Fig. 4. As Thm. 1 states, CV reaches the same testing accuracy as Exact, while NS and NS+PP perform much worse. Testing using exact algorithms (CV or Exact) corresponds to the weight scaling algorithm for dropout (Srivastava et al., 2014).

Finally, we compare the average bias and variance of the gradients per dimension for first layer weights relative to the weights' magnitude in Fig. 5. For models without dropout, the gradient of CV+PP is almost unbiased. For models with dropout, the bias and variance of CV+PP and CVD+PP are ususaly smaller than NS and NS+PP, as we analyzed in Sec. 4.3.

## 6 CONCLUSIONS

The large receptive field size of GCN hinders its fast stochastic training. In this paper, we present a preprocessing strategy and two control variate based algorithms to reduce the receptive field size. Our algorithms can achieve comparable convergence speed with the exact algorithm even the neighbor sampling size $D^{(l)} = 2$, so that the per-epoch cost of training GCN is comparable with training MLPs. We also present strong theoretical guarantees, including exact prediction and convergence to GCN's local optimum, for our control variate based algorithm.

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

# A    PROOF OF THEOREM 1

*Proof.* 1. We prove by induction. After the first epoch the activation $h_{i,v}^{(0)}$ is at least computed once for each node $v$, so $\bar{H}_{CV,i}^{(0)} = H_{CV,i}^{(0)} = H^{(0)}$ for all $i > I$. Assume that we have $\bar{H}_{CV,i}^{(l)} = H_{CV,i}^{(l)} = H^{(l)}$ for all $i > (l+1)I$. Then for all $i > (l+1)I$

$$Z_{CV,i}^{(l+1)} = \left( \hat{P}_i^{(l)}(H_{CV,i}^{(l)} - \bar{H}_{CV,i}^{(l)}) + P\bar{H}_{CV,i}^{(l)} \right) W^{(l)} = P\bar{H}_{CV,i}^{(l)}W^{(l)} = PH^{(l)}W^{(l)} = Z^{(l+1)}. \tag{7}$$

$$H_{CV,i}^{(l+1)} = \sigma(Z_{CV,i}^{(l+1)}) = H^{(l+1)}$$

After one more epoch, all the activations $h_{CV,i,v}^{(l+1)}$ are computed at least once for each $v$, so $\bar{H}_{CV,i}^{(l+1)} = H_{CV,i}^{(l+1)} = H^{(l+1)}$ for all $i > (l+2)I$. By induction, we know that after $LI$ steps, we have $\bar{H}_{CV,i}^{(L-1)} = H_{CV,i}^{(L-1)} = H^{(L-1)}$. By Eq. 7 we have $\bar{Z}_{CV,i}^{(L)} = Z^{(L)}$.

2. We omit the time subscript $_i$ and denote $f_{CV,v} := f(y_v, z_{CV,v}^{(L)})$. By back propagation, the approximated gradients by CV can be computed as follows

$$\nabla_{H_{CV}^{(l)}} f_{CV,v} = \hat{P}^{(l)}\nabla_{Z_{CV}^{(l+1)}} f_{CV,v} W^{(l)\top} \qquad l = 1, \ldots, L-1$$

$$\nabla_{Z_{CV}^{(l)}} f_{CV,v} = \sigma'(Z_{CV}^{(l)}) \circ \nabla_{H_{CV}^{(l)}} f_{CV,v} \qquad l = 1, \ldots, L-1$$

$$\nabla_{W^{(l)}} f_{CV,v} = (\hat{P}^{(l)} H_{CV}^{(l)})^\top \nabla_{Z_{CV}^{(l+1)}} f_{CV,v} \qquad l = 0, \ldots, L-1,$$

$$g_{CV}(W) = \frac{1}{|\mathcal{V}_B|} \sum_{v \in \mathcal{V}_B} \nabla_W f_{CV,v}, \tag{8}$$

where $\circ$ is the element wise product and $\sigma'(Z_{CV}^{(l)})$ is the element-wise derivative. Similarly, denote $f_v := f(y_v, Z_v^{(l)})$, the exact gradients can be computed as follows

$$\nabla_{H^{(l)}} f_v = P^\top \nabla_{Z^{(l+1)}} f_v W^{(l)\top} \qquad l = 1, \ldots, L-1$$

$$\nabla_{Z^{(l)}} f_v = \sigma'(Z^{(l)}) \circ \nabla_{H^{(l)}} f_v \qquad l = 1, \ldots, L-1$$

$$\nabla_{W^{(l)}} f_v = (PH^{(l)})^\top \nabla_{Z^{(l+1)}} f_v \qquad l = 0, \ldots, L-1,$$

$$g(W) = \frac{1}{V} \sum_{v \in \mathcal{V}} \nabla_W f_v. \tag{9}$$

Applying $\mathbb{E}_{\hat{P}} = \mathbb{E}_{\hat{P}^{(1)}, \ldots, \hat{P}^{(L)}}$ to both sides of Eq. 8, and utilizing

- 1's conclusion that after $L$ epoches, $Z_{CV}^{(l)} = Z^{(l)}$, so $\nabla_{Z_{CV}^{(L)}} f_{CV,v}$ is also determinstic.

- $\mathbb{E}_{\hat{P}}[\nabla_{Z^{(l)}} f_{CV,v}] = \mathbb{E}_{\hat{P}^{(l)}, \ldots, \hat{P}^{(L)}}[\nabla_{Z^{(l)}} f_{CV,v}]$.

- $\mathbb{E}_{\hat{P}}[\nabla_{H^{(l)}} f_{CV,v}] = \mathbb{E}_{\hat{P}^{(l)}, \ldots, \hat{P}^{(L)}}[\nabla_{H^{(l)}} f_{CV,v}]$.

we have

$$\mathbb{E}_{\hat{P}^{(l)}, \ldots, \hat{P}^{(L)}} \nabla_{H_{CV}^{(l)}} f_{CV,v} = \mathbb{E}_{\hat{P}^{(l)}} \hat{P}^{(l)\top} \mathbb{E}_{\hat{P}^{(l+1)}, \ldots, \hat{P}^{(L)}}[\nabla_{Z_{CV}^{(l+1)}} f_{CV,v}] W^{(l)\top} \quad l = 1, \ldots, L-1$$

$$\mathbb{E}_{\hat{P}^{(l)}, \ldots, \hat{P}^{(L)}} \nabla_{Z_{CV}^{(l)}} f_{CV,v} = \sigma'(Z_{CV}^{(l)}) \circ \mathbb{E}_{\hat{P}^{(l)}, \ldots, \hat{P}^{(L)}} \nabla_{H_{CV}^{(l)}} f_{CV,v} \quad l = 1, \ldots, L-1$$

$$\mathbb{E}_{\hat{P}} \nabla_{W^{(l)}} f_{CV,v} = H^{(l)\top} \mathbb{E}_{\hat{P}^{(l)}} \hat{P}^{(l)\top} \mathbb{E}_{\hat{P}^{(l+1)}, \ldots, \hat{P}^{(L)}} \nabla_{Z_{CV}^{(l+1)}} f_{CV,v} \quad l = 0, \ldots, L-1,$$

$$g_{CV}(W) = \frac{1}{|\mathcal{V}_B|} \sum_{v \in \mathcal{V}_B} \mathbb{E}_{\hat{P}} \nabla_W f_{CV,v}. \tag{10}$$

Comparing Eq. 10 and Eq. 9 we get

$$\mathbb{E}_{\hat{P}} \nabla_{W^{(l)}} f_{CV,v} = \nabla_{W^{(l)}} f_v, \quad l = 0, \ldots, L-1,$$

so

$$\mathbb{E}_{\hat{P},\mathcal{V}_B} g_{CV}(W) = \mathbb{E}_{\mathcal{V}_B} \frac{1}{|\mathcal{V}_B|} \sum_{v \in \mathcal{V}_B} \mathbb{E}_{\hat{P}} \nabla_W f_{CV,v} = \frac{1}{V} \sum_{v \in \mathcal{V}} \nabla_W f_v.$$

$\square$

# B   PROOF OF THEOREM 2

We proof Theorem 2 in 3 steps:

1. Lemma 1: For a sequence of weights $W^{(1)}, \dots, W^{(N)}$ which are close to each other, CV's approximate activations are close to the exact activations.

2. Lemma 2: For a sequence of weights $W^{(1)}, \dots, W^{(N)}$ which are close to each other, CV's gradients are close to be unbiased.

3. Theorem 2: An SGD algorithm generates the weights that changes slow enough for the gradient bias goes to zero, so the algorithm converges.

The following proposition is needed in our proof

**Proposition 1.** *Let* $\|A\|_\infty = \max_{ij} |A_{ij}|$, *then*

- $\|AB\|_\infty \le col(A) \|A\|_\infty \|B\|_\infty$, *where* $col(A)$ *is the number of columns of the matrix A.*

- $\|A \circ B\|_\infty \le \|A\|_\infty \|B\|_\infty$.

- $\|A + B\|_\infty \le \|A\|_\infty + \|B\|_\infty$.

*Proof.*

$$\|AB\|_\infty = \max_{ij} \left| \sum_k A_{ik} B_{ik} \right| \le \max_{ij} \left| \sum_k \|A\|_\infty \|B\|_\infty \right| = col(A) \|A\|_\infty \|B\|_\infty.$$

$$\|A \circ B\|_\infty = \max_{ij} |A_{ij} B_{ij}| \le \max_{ij} \|A\|_\infty \|B\|_\infty = \|A\|_\infty \|B\|_\infty.$$

$$\|A + B\|_\infty = \max_{ij} |A_{ij} + B_{ij}| \le \max_{ij} \{|A_{ij}| + |B_{ij}|\} \le \max_{ij} |A_{ij}| + \max_{ij} |B_{ij}| = \|A\|_\infty + \|B\|_\infty.$$

$\square$

We define $C := \max\{col(P), col(H^{(0)}), \dots, col(H^{(L)})\}$.

## B.1   PROOF OF LEMMA 1

**Proposition 2.** *There are a series of T inputs* $X_1, \dots, X_T$, $X_{CV,1}, \dots, X_{CV,T}$ *and weights* $W_1, \dots, W_T$ *feed to an one-layer GCN with CV*

$$Z_{CV,i} = \left( \hat{P}_i(X_i - \bar{X}_i) + P\bar{X}_i \right) W_i, \quad H_{CV,i} = \sigma(Z_{CV,i}), \quad \bar{H}_{CV,i+1} = \mathbf{s}_i H_{CV,i} + (1 - \mathbf{s}_i) \bar{H}_{CV,i}.$$

*and an one-layer exact GCN*

$$Z_i = P X_i W_i, \quad H_i = \sigma(Z_i).$$

*If*

1. *The activation* $\sigma(\cdot)$ *is* $\rho$*-Lipschitz;*

2. $\|X_{CV,i} - X_{CV,j}\|_\infty < \epsilon$ *and* $\|X_{CV,i} - X_i\|_\infty < \epsilon$ *for all* $i, j \le T$ *and* $\epsilon > 0$.

*Then there exists some* $K > 0$*, s.t.,* $\|H_{CV,i} - H_{CV,j}\|_\infty < K\epsilon$ *and* $\|H_{CV,i} - H_i\|_\infty < K\epsilon$ *for all* $I < i, j \le T$*, where I is the number of iterations per epoch.*

*Proof.* Because for all $i > I$, the elements of $\bar{X}_{CV,i}$ are all taken from previous epochs, i.e., $X_{CV,1}, \ldots, X_{CV,i-1}$, we know that

$$\left\| \bar{X}_{CV,i} - X_{CV,i} \right\|_\infty \le \max_{j \le i} \left\| X_{CV,j} - X_{CV,i} \right\|_\infty \le \epsilon \quad (\forall i > I). \tag{11}$$

By triangular inequality, we also know

$$\left\| \bar{X}_{CV,i} - \bar{X}_{CV,j} \right\|_\infty < 3\epsilon \quad (\forall i, j > I). \tag{12}$$

$$\left\| \bar{X}_{CV,i} - X_i \right\|_\infty < 2\epsilon \quad (\forall i > I). \tag{13}$$

Since $\|X_{CV,1}\|_\infty, \ldots, \|X_{CV,T}\|_\infty$ are bounded, $\left\| \bar{X}_{CV,i} \right\|_\infty$ is also bounded for $i > I$. Then,

$$
\begin{aligned}
\|H_{CV,i} - H_{CV,j}\|_\infty &\le \rho \|Z_{CV,i} - Z_{CV,j}\|_\infty \\
&\le \rho \left\| \left( \hat{P}_i(X_{CV,i} - \bar{X}_{CV,i}) + P\bar{X}_{CV,i} \right) W_i - \left( \hat{P}_j(X_{CV,j} - \bar{X}_{CV,j}) + P\bar{X}_{CV,j} \right) W_j \right\|_\infty \\
&\le \rho \left\| \hat{P}_i(X_{CV,i} - \bar{X}_{CV,i}) W_i - \hat{P}_j(X_{CV,j} - \bar{X}_{CV,j}) W_j \right\|_\infty + \rho \left\| P\bar{X}_{CV,i}W_i - P\bar{X}_{CV,j}W_j \right\|_\infty \\
&\le \rho C^2 [ \left\| \hat{P}_i - \hat{P}_j \right\|_\infty \|X_{CV,i} - \bar{X}_{CV,i}\|_\infty \|W_i\|_\infty \\
&\quad + \left\| \hat{P}_j \right\|_\infty \|X_{CV,i} - \bar{X}_{CV,i} - X_{CV,j} + \bar{X}_{CV,j}\|_\infty \|W_i\|_\infty \\
&\quad + \left\| \hat{P}_j \right\|_\infty \|X_{CV,j} - \bar{X}_{CV,j}\|_\infty \|W_i - W_j\|_\infty \\
&\quad + \|P\|_\infty \|\bar{X}_{CV,i} - \bar{X}_{CV,j}\|_\infty \|W_i\|_\infty + \|P\|_\infty \|\bar{X}_{CV,j}\|_\infty \|W_i - W_j\|_\infty ] \\
&\le \rho C^2 \epsilon [ \left\| \hat{P}_i - \hat{P}_j \right\|_\infty \|W_i\|_\infty + 2 \left\| \hat{P}_j \right\|_\infty \|W_i\|_\infty + \left\| \hat{P}_j \right\|_\infty \|W_i - W_j\|_\infty + \\
&\quad 3 \left\| \hat{P}_j \right\|_\infty \|W_i\|_\infty + \left\| \hat{P}_j \right\|_\infty \|\bar{X}_{CV,j}\|_\infty ] \\
&= K_1 \epsilon,
\end{aligned}
$$

and

$$
\begin{aligned}
\|H_{CV,i} - H_i\|_\infty &\le \rho \|Z_{CV,i} - Z_i\|_\infty \\
&\le \rho \left\| \left( \hat{P}_i(X_{CV,i} - \bar{X}_{CV,i}) + P(\bar{X}_{CV,i} - X_i) \right) \right\|_\infty W_i \\
&\le \rho C ( \left\| \hat{P}_i \right\|_\infty \epsilon + 2 \|P\|_\infty \epsilon) \|W_i\|_\infty \\
&\le K_2 \epsilon.
\end{aligned}
$$

$\square$

The following lemma bounds CV's approximation error of activations

**Lemma 1.** *Given a sequence of model weights* $W_1, \ldots, W_T$. *If* $\|W_i - W_j\|_\infty < \epsilon, \forall i, j$, *and all the activations are* $\rho$*-Lipschitz, there exists* $K > 0$, *s.t.,*

- $\left\| H_i^l - H_{CV,i}^l \right\|_\infty < K\epsilon, \forall i > LI, l = 1, \ldots, L - 1$,

- $\left\| Z_i^l - Z_{CV,i}^l \right\|_\infty < K\epsilon, \forall i > LI, l = 1, \ldots, L$.

*Proof.* We prove by induction. Because $H^0 = X$ is constant, $\bar{H}_{CV,i}^0 = H_i^0$ after $I$ iterations. So $H_{CV,i}^1 = \sigma( \left( \hat{P}_i(H_{CV,i}^0 - \bar{H}_{CV,i}^0) + P\bar{H}_{CV,i}^0 \right) W_i^0 = \sigma(PXW_i^0) = H_i^1$, and

$$\left\| H_{CV,i}^1 - H_{CV,j}^1 \right\|_\infty = \left\| \sigma(PXW_i^0) - \sigma(PXW_j^0) \right\|_\infty \le \rho C \|P\|_\infty \|X\|_\infty \epsilon.$$

Repeatedly apply Proposition B.1 for $L - 1$ times, we get the intended results. $\square$

## B.2 PROOF OF LEMMA 2

The following lemma bounds the bias of CV's approximate gradient

**Lemma 2.** *Given a sequence of model weights $W_1, \ldots, W_T$, if*

1. $\|W_i - W_j\|_\infty < \epsilon, \forall i, j,$

2. *all the activations are $\rho$-Lipschitz,*

3. *the gradient of the cost function $\nabla_z f(y, z)$ is $\rho$-Lipschitz and bounded,*

*then there exists $K > 0$, s.t.,*

$$\left\| \mathbb{E}_{\hat{P}, \mathcal{V}_B} g_{CV}(W_i) - g(W_i) \right\|_\infty < K\epsilon, \forall i > LI.$$

*Proof.* By Lipschitz continuity of $\nabla_z f(y, z)$ and Lemma 1, there exists $K > 0$, s.t.,

$$\left\| \nabla_{Z_{CV}^{(l)}} f_{CV,v} - \nabla_{Z^{(l)}} f_v \right\|_\infty \leq \rho \left\| Z_{CV}^{(l)} - Z^{(l)} \right\|_\infty \leq \rho K \epsilon.$$

Assume that $\left\| \mathbb{E}_{\hat{P}} \nabla_{Z_{CV}^{(l+1)}} f_{CV,v} - \nabla_{Z^{(l+1)}} f_v \right\|_\infty < K_1 \epsilon$, we now prove that there exists $K > 0$, s.t., $\left\| \mathbb{E}_{\hat{P}} \nabla_{Z_{CV}^{(l)}} f_{CV,v} - \nabla_{Z^{(l)}} f_v \right\|_\infty < K\epsilon$. By Eq. 9, Eq. 10 and Lemma 1, we have

$$\left\| \mathbb{E}_{\hat{P}} \nabla_{H_{CV}^{(l)}} f_{CV,v} - \nabla_{H^{(l)}} f_v \right\|_\infty$$
$$= \left\| P^\top \mathbb{E}_{\hat{P}} [\nabla_{Z_{CV}^{(l+1)}} f_{CV,v}] W^{(l)\top} - P^\top \nabla_{Z^{(l+1)}} f_v W^{(l)\top} \right\|_\infty$$
$$\leq \left\| P^\top \right\|_\infty K_1 C^2 \epsilon \left\| W^{(l)\top} \right\|_\infty,$$

and

$$\left\| \mathbb{E}_{\hat{P}} \nabla_{Z_{CV}^{(l)}} f_{CV,v} - \nabla_{Z^{(l)}} f_v \right\|_\infty$$
$$= \left\| \mathbb{E}_{\hat{P}} \left[ \sigma'(Z_{CV}^{(l)}) \circ \nabla_{H_{CV}^{(l)}} f_{CV,v} \right] - \sigma'(Z^{(l)}) \circ \nabla_{H^{(l)}} f_v \right\|_\infty$$
$$\leq \left\| \mathbb{E}_{\hat{P}} \left[ (\sigma'(Z_{CV}^{(l)}) - \sigma'(Z^{(l)})) \circ \nabla_{H_{CV}^{(l)}} f_{CV,v} \right] \right\|_\infty + \left\| \sigma'(Z^{(l)})(\mathbb{E}_{\hat{P}} [\nabla_{H_{CV}^{(l)}} f_{CV,v}] - \nabla_{H^{(l)}} f_v) \right\|_\infty$$
$$\leq \left\| \mathbb{E}_{\hat{P}} \left[ \rho K C \epsilon \circ \nabla_{H_{CV}^{(l)}} f_{CV,v} \right] \right\|_\infty + \left\| \sigma'(Z^{(l)}) \right\|_\infty \left\| P^\top \right\|_\infty K_1 C^3 \epsilon \left\| W^{(l)\top} \right\|_\infty$$
$$\leq \rho K C^2 \epsilon \left\| \mathbb{E}_{\hat{P}} \nabla_{H_{CV}^{(l)}} f_{CV,v} \right\|_\infty + \left\| \sigma'(Z^{(l)}) \right\|_\infty \left\| P^\top \right\|_\infty K_1 C^3 \epsilon \left\| W^{(l)\top} \right\|_\infty \leq K_2 \epsilon$$

By induction we know that for $l = 1, \ldots, L$ there exists $K$, s.t.,

$$\left\| \mathbb{E}_{\hat{P}} \nabla_{Z_{CV}^{(l)}} f_{CV,v} - \nabla_{Z^{(l)}} f_v \right\|_\infty \leq K\epsilon.$$

Again by Eq. 9, Eq. 10, and Lemma 1,

$$\left\| \mathbb{E}_{\hat{P}} \nabla_{W^{(l)}} f_{CV,v} - \nabla_{W^{(l)}} f_v \right\|_\infty$$
$$= \left\| \mathbb{E}_{\hat{P}} \left[ H_{CV}^{(l)\top} P^\top \nabla_{Z_{CV}^{(l)}} f_{CV,v} \right] - H^{(l)\top} P^\top \nabla_{Z^{(l)}} f_v \right\|_\infty$$
$$\leq \left\| \mathbb{E}_{\hat{P}} \left[ \left( H_{CV}^{(l)\top} - H^{(l)\top} \right) P^\top \nabla_{Z_{CV}^{(l)}} f_{CV,v} \right] \right\|_\infty + \left\| H^{(l)\top} P^\top \left( \mathbb{E}_{\hat{P}} \nabla_{Z_{CV}^{(l)}} f_{CV,v} - \nabla_{Z^{(l)}} f_v \right) \right\|_\infty$$
$$\leq \left\| \mathbb{E}_{\hat{P}} \left[ K C \epsilon P^\top \nabla_{Z_{CV}^{(l)}} f_{CV,v} \right] \right\|_\infty + C^2 \left\| H^{(l)\top} \right\|_\infty \left\| P^\top \right\|_\infty K\epsilon$$
$$\leq K C^2 \epsilon \left\| P \right\|_\infty \left\| \mathbb{E}_{\hat{P}} \left[ \nabla_{Z_{CV}^{(l)}} f_{CV,v} \right] \right\|_\infty + \left\| H^{(l)\top} \right\|_\infty \left\| P^\top \right\|_\infty K C^2 \epsilon \leq K_3 \epsilon$$

Finally,

$$\left\| \mathbb{E}_{\hat{P}, \mathcal{V}_B} g_{CV}(W_i) - g(W_i) \right\|_\infty$$

$$= \left\| \mathbb{E}_{\mathcal{V}_B} \left( \frac{1}{|\mathcal{V}_B|} \sum_{v \in \mathcal{V}_B} \mathbb{E}_{\hat{P}} \left[ \nabla_{W^{(l)}} f_{CV,v} \right] - \frac{1}{V} \sum_{v \in \mathcal{V}} \nabla_{W^{(l)}} f_v \right) \right\|_\infty$$

$$= \left\| \frac{1}{V} \sum_{v \in \mathcal{V}} \left( \mathbb{E}_{\hat{P}} \left[ \nabla_{W^{(l)}} f_{CV,v} \right] - \nabla_{W^{(l)}} f_v \right) \right\|_\infty \leq K_3 \epsilon.$$

$\square$

### B.3 Proof of Theorem 2

*Proof.* This proof is a modification of Ghadimi & Lan (2013), but using biased stochastic gradients instead. We assume the algorithm is already warmed-up for $LI$ steps with the initial weights $W_0$, so that Lemma 2 holds for step $i > 0$. Denote $\delta_i = g_{CV}(W_i) - \nabla \mathcal{L}(W_i)$. By smoothness we have

$$\mathcal{L}(W_{i+1}) \leq \mathcal{L}(W_i) + \langle \nabla \mathcal{L}(W_i), W_{i+1} - W_i \rangle + \frac{\rho}{2} \gamma_i^2 \| g_{CV}(W_i) \|^2$$

$$= \mathcal{L}(W_i) - \gamma_i \langle \nabla \mathcal{L}(W_i), g_{CV}(W_i) \rangle + \frac{\rho}{2} \gamma_i^2 \| g_{CV}(W_i) \|^2$$

$$= \mathcal{L}(W_i) - \gamma_i \langle \nabla \mathcal{L}(W_i), \delta_i \rangle - \gamma_i \| \nabla \mathcal{L}(W_i) \|^2 + \frac{\rho}{2} \gamma_i^2 \left[ \| \delta_i \|^2 + \| \nabla \mathcal{L}(W_i) \|^2 + 2 \langle \delta_i, \nabla \mathcal{L}(W_i) \rangle \right]$$

$$= \mathcal{L}(W_i) - (\gamma_i - \rho \gamma_i^2) \langle \nabla \mathcal{L}(W_i), \delta_i \rangle - (\gamma_i - \frac{\rho \gamma_i^2}{2}) \| \nabla \mathcal{L}(W_i) \|^2 + \frac{\rho}{2} \gamma_i^2 \| \delta_i \|^2. \qquad (14)$$

Consider the sequence of $LI + 1$ weights $W_{i-LI}, \ldots, W_i$.

$$\max_{i-LI \leq j, k \leq i} \| W_j - W_k \|_\infty \leq \sum_{j=i-LI}^{i-1} \| W_j - W_{j+1} \|_\infty$$

$$= \sum_{j=i-LI}^{i-1} \gamma_j \| g_{CV}(W_j) \|_\infty \leq \sum_{j=i-LI}^{i-1} \gamma_j G \leq LIG\gamma_{i-LI}.$$

By Lemma 2, there exists $K > 0$, s.t.

$$\mathbb{E}_{\hat{P}, \mathcal{V}_B} \| \delta_i \|_\infty = \mathbb{E}_{\hat{P}, \mathcal{V}_B} \| g_{CV}(W_i) - \nabla \mathcal{L}(W_i) \|_\infty \leq KLIG\gamma_{i-LI}, \quad \forall i > 0.$$

Assume that $W$ is $D$-dimensional,

$$\mathbb{E}_{\hat{P}, \mathcal{V}_B} \langle \nabla \mathcal{L}(W_i), \delta_i \rangle \geq -\mathbb{E}_{\hat{P}, \mathcal{V}_B} D \| \nabla \mathcal{L}(W_i) \|_\infty \| \delta_i \|_\infty \geq -KLIDG^2 \gamma_{i-LI} = K_1 \gamma_{i-LI},$$

$$\mathbb{E}_{\hat{P}, \mathcal{V}_B} \| \delta_i \|^2 \leq D \left( \mathbb{E}_{\hat{P}, \mathcal{V}_B} \| \delta_i \|_\infty \right)^2 \leq DK^2 L^2 B^2 G^2 \gamma_{i-LI} = K_2 \gamma_{i-LI},$$

where $K_1 = KLIDG^2$ and $K_2 = DK^2 L^2 B^2 G^2$. Taking $\mathbb{E}_{\hat{P}, \mathcal{V}_B}$ to both sides of Eq. 14 we have

$$\mathcal{L}(W_{i+1}) \leq \mathcal{L}(W_i) + (\gamma_i - \rho \gamma_i^2) K_1 \gamma_{i-LI} - (\gamma_i - \frac{\rho \gamma_i^2}{2}) \mathbb{E}_{\hat{P}, \mathcal{V}_B} \| \nabla \mathcal{L}(W_i) \|^2 + \frac{\rho}{2} \gamma_i^2 K_2 \gamma_{i-LI}.$$

Summing up the above inequalities and re-arranging the terms, we obtain,

$$\sum_{i=1}^{N} (\gamma_i - \frac{\rho \gamma_i^2}{2}) \mathbb{E}_{\hat{P}, \mathcal{V}_B} \| \nabla \mathcal{L}(W_i) \|^2$$

$$\leq \mathcal{L}(W_1) - \mathcal{L}(W^*) + K_1 \sum_{i=1}^{N} (\gamma_i - \rho \gamma_i^2) \gamma_{i-LI} + \frac{\rho K_2}{2} \sum_{i=1}^{N} \gamma_i^2 \gamma_{i-LI}.$$

Dividing both sides by $\sum_{i=1}^{N}(\gamma_i - \frac{\rho\gamma_i^2}{2})$,

$$\mathbb{E}_{R\sim P_R}\mathbb{E}_{\hat{P},\mathcal{V}_B}\left\|\nabla\mathcal{L}(W_R)\right\|^2$$
$$\leq 2\frac{\mathcal{L}(W_1) - \mathcal{L}(W^*) + K_1\sum_{i=1}^{N}(\gamma_i - \rho\gamma_i^2)\gamma_{i-LI} + \frac{\rho K_2}{2}\sum_{i=1}^{N}\gamma_i^2\gamma_{i-LI}}{\sum_{i=1}^{N}\gamma_i(2 - \rho\gamma_i)}.$$

Taking $\gamma_i = \gamma := \min\{\frac{1}{\rho}, \frac{1}{\sqrt{N}}\}$, for all $i = 1, \ldots, N$, we have

$$\mathbb{E}_{R\sim P_R}\mathbb{E}_{\hat{P},\mathcal{V}_B}\left\|\nabla\mathcal{L}(W_R)\right\|^2$$
$$\leq 2\frac{\mathcal{L}(W_1) - \mathcal{L}(W^*) + K_1 N(\gamma - \rho\gamma^2)\gamma + \frac{\rho K_2}{2}N\gamma^3}{N\gamma(2 - \rho\gamma)}$$
$$\leq 2\frac{\mathcal{L}(W_1) - \mathcal{L}(W^*) + K_1 N(\gamma - \rho\gamma^2)\gamma + \frac{\rho K_2}{2}N\gamma^3}{N\gamma}$$
$$\leq 2\frac{\mathcal{L}(W_1) - \mathcal{L}(W^*)}{N\gamma} + K_1\gamma(1 - \rho\gamma) + \rho K_2\gamma^2$$
$$\leq \frac{2\rho(\mathcal{L}(W_1) - \mathcal{L}(W^*)) + K_2}{N} + \frac{2(\mathcal{L}(W_1) - \mathcal{L}(W^*)) + K_1}{\sqrt{N}}.$$

Particularly, when $N \to \infty$, we have $\mathbb{E}_{R\sim P_R}\mathbb{E}_{\hat{P},\mathcal{V}_B}\left\|\nabla\mathcal{L}(W_R)\right\|^2 = 0$, which implies that the gradient is asymptotically unbiased.

$\square$

## C    DERIVATION OF THE VARIANCE

$$\text{Var}[u] = \mathbb{E}\left[\sum_v p_v(h_v - \mu_v)\right]^2$$
$$= \sum_v p_v^2\mathbb{E}\left[(h_v - \mu_v)\right]^2$$
$$= \sum_v p_v^2 s_v^2$$
$$= s^2.$$

$$\text{Var}[u_{NS}] = \mathbb{E}\left[Dp_{v'}h_{v'} - \mu\right]^2$$
$$= \mathbb{E}_{v'}\{D^2 p_{v'}^2(\mu_{v'}^2 + s_{v'}^2) + \mu^2 - D\mu p_{v'}\mu_{vp}\}$$
$$= Ds^2 + (D\sum_v p_v^2\mu_v^2 - \mu^2)$$
$$= Ds^2 + \frac{1}{2}\sum_{v,v'}(p_v\mu_v - p_{v'}\mu_{v'})^2.$$

$$\text{Var}[u_{CV}] = \mathbb{E}\left[ Dp_{v'}\Delta h_{v'} + \sum_v p_v(\bar{h}_v - \mu_v) \right]^2$$

$$= \mathbb{E}\left[ Dp_{v'}\Delta h_{v'} + \sum_v p_v(\bar{h}_v - \bar{\mu}_v) - \Delta\mu \right]^2$$

$$= \mathbb{E}_{v'}\left\{ D^2 p_{v'}^2 \mathbb{E}\left(\Delta h_{v'}\right)^2 + \sum_v p_v^2 \mathbb{E}\left(\bar{h}_v - \bar{\mu}_v\right)^2 + \Delta\mu^2 + 2Dp_{v'}\sum_v p_v \mathbb{E}\left[\Delta h_{v'}(\bar{h}_v - \bar{\mu}_v)\right] \right\}$$

$$\quad - \mathbb{E}_{v'}\left\{ 2Dp_{v'}\Delta\mu\mathbb{E}\Delta h_{v'} - 2\Delta\mu\sum_v p_v \mathbb{E}(\bar{h}_v - \bar{\mu}_v) \right\}$$

$$= \mathbb{E}_{v'}\left\{ D^2 p_{v'}^2 (\Delta\mu_{v'}^2 + \Delta s_{v'}^2) + \sum_v p_v^2 \bar{s}_v^2 + \Delta\mu^2 - 2Dp_{v'}^2 \bar{s}_{v'}^2 - 2Dp_{v'}\Delta\mu\Delta\mu_{v'} \right\}$$

$$= D\sum_v p_v^2 \Delta\mu_v^2 + D\Delta s^2 + \bar{s}^2 + \Delta\mu^2 - 2\bar{s}^2 - 2\Delta\mu^2$$

$$= (D\Delta s^2 - \bar{s}^2) + (D\sum_v p_v^2 \Delta\mu_v^2 - \Delta\mu^2)$$

$$= \left[ Ds^2 + (D-1)\bar{s}^2 \right] + \frac{1}{2}\sum_{v,v'}(p_v\Delta\mu_v - p_{v'}\Delta\mu_{v'})^2.$$

$$\text{Var}[u_{CVD}] = \mathbb{E}\left[ \sqrt{D}p_{v'}(h_{v'} - \mu_{v'}) + Dp_{v'}\Delta\mu_{v'} + \sum_v p_v(\bar{\mu}_v - \mu_v) \right]^2$$

$$= \mathbb{E}\left[ \sqrt{D}p_{v'}(h_{v'} - \mu_{v'}) + Dp_{v'}\Delta\mu_{v'} - \Delta\mu \right]^2$$

$$= \mathbb{E}_{v'}\left\{ Dp_{v'}^2 \mathbb{E}(h_{v'} - \mu_{v'})^2 + D^2 p_{v'}^2 \Delta\mu_{v'}^2 + \Delta\mu^2 - 2Dp_{v'}\Delta\mu_{v'}\Delta\mu \right\}$$

$$= \sum_v p_v^2 s_v^2 + D\sum_v p_v^2 \Delta\mu_v^2 + \Delta\mu^2 - 2\Delta\mu^2$$

$$= s^2 + (D\sum_v p_v^2 \Delta\mu_v^2 - \Delta\mu^2)$$

$$= s^2 + \frac{1}{2}\sum_{v,v'}(p_v\Delta\mu_v - p_{v'}\Delta\mu_{v'})^2.$$

## D  EXPERIMENT SETUP

In this sections we describe the details of our model architectures. We use the Adam optimizer Kingma & Ba (2014) with learning rate 0.01.

- Citeseer, Cora, PubMed and NELL: We use the same architecture as Kipf & Welling (2017): two graph convolution layers with one linear layer per graph convolution layer. We use 32 hidden units, 50% dropout rate and $5 \times 10^{-4}$ L2 weight decay for Citeseer, Cora and PubMed and 64 hidden units, 10% dropout rate and $10^{-5}$ L2 weight decay for NELL.

- PPI and Reddit: We use the mean pooling architecture proposed by Hamilton et al. (2017a). We use two linear layers per graph convolution layer. We set weight decay to be zero, dropout rate to be 0.2%, and adopt layer normalization (Ba et al., 2016) after each linear layer. We use 512 hidden units for PPI and 128 hidden units for Reddit. We find that our architecture can reach 97.8% testing micro-F1 on the PPI dataset, which is significantly higher than 59.8% reported by Hamilton et al. (2017a). We find the improvement is from wider hidden layer, dropout and layer normalization.

---

**Algorithm 1** Training with the CV algorithm

---

**for** each minibatch $\mathcal{V}_B \subset \mathcal{V}$ **do**
    Randomly sample propagation matrices $\hat{P}^{(0)}, \ldots, \hat{P}^{(L-1)}$
    Compute the receptive fields $\mathbf{m}^{(0)}, \ldots, \mathbf{m}^{(L-1)}$
    (Forward propgation)
    **for** each layer $l \leftarrow 0$ to $L-1$ **do**
        $Z^{(l+1)} \leftarrow \left( \hat{P}^{(l)}(H^{(l)} - \bar{H}^{(l)} + P\bar{H}^{(l)}) \right) W^{(l)}$
        $H^{(l+1)} \leftarrow \sigma(Z^{(l+1)})$
    **end for**
    Compute the loss $\mathcal{L} = \frac{1}{|\mathcal{V}_B|} \sum_{v \in \mathcal{V}_B} f(y_v, Z_v^{(L)})$
    (Backward propagation)
    $W \leftarrow W - \gamma_i \nabla_W \mathcal{L}$
    (Update historical activations)
    **for** each layer $l \leftarrow 0$ to $L-1$ **do**
        $\bar{H}^{(l)} \leftarrow \mathbf{m}^{(l)} H^{(l)} + (1 - \mathbf{m}^{(l)})\bar{H}^{(l)}$
    **end for**
**end for**

---

**Algorithm 2** Training with the CVD algorithm

---

**for** each minibatch $\mathcal{V}_B \subset \mathcal{V}$ **do**
    Randomly sample propagation matrices $\hat{P}^{(0)}, \ldots, \hat{P}^{(L-1)}$
    Compute the receptive fields $\mathbf{m}^{(0)}, \ldots, \mathbf{m}^{(L-1)}$
    (Forward propgation)
    **for** each layer $l \leftarrow 0$ to $L-1$ **do**
        $U \leftarrow \left( \bar{P}^{(l)}(H^{(l)} - \mu^{(l)}) + \hat{P}^{(l)}(\mu^{(l)} - \bar{\mu}^{(l)}) + P\bar{H}^{(l)} \right)$
        $H^{(l+1)} \leftarrow \sigma(\text{Dropout}_p(U)W^{(l)})$
        $\mu^{(l+1)} \leftarrow \sigma(UW^{(l)})$
    **end for**
    Compute the loss $\mathcal{L} = \frac{1}{|\mathcal{V}_B|} \sum_{v \in \mathcal{V}_B} f(y_v, H_v^{(L)})$
    (Backward propagation)
    $W \leftarrow W - \gamma_i \nabla_W \mathcal{L}$
    (Update historical activations)
    **for** each layer $l \leftarrow 0$ to $L-1$ **do**
        $\bar{\mu}^{(l)} \leftarrow \mathbf{m}^{(l)} \mu^{(l)} + (1 - \mathbf{m}^{(l)})\bar{\mu}^{(l)}$
    **end for**
**end for**

---

# E    PSEUDOCODE

## E.1    TRAINING WITH THE CV ESTIMATOR

Alg. 1 depicts the training algorithm using the CV estimator in Sec. 4.1. We perform forward propagation according to Eq. (5,6), compute the stochastic gradient, and then update the historical activations $\bar{H}^{(l)}$ according to Eq. (6). We omit the subscripts $CV$ and the iteration number $i$ for concise. Let $W = (W^{(0)}, \ldots, W^{(L-1)})$ be all the trainable parameters, the gradient $\nabla_W \mathcal{L}$ is computed automatically by frameworks such as TensorFlow. The diagonal matrix $\mathbf{m}^{(l)}$ denotes the receptive field at layer $l$, i.e., the nodes that need to be computed in order to approximate $z_v^{(l)}$ for $v$ in the minibatch $\mathcal{V}_B$. We only need to compute and update the activations $H^{(l)}$, $Z^{(l)}$, $\bar{H}^{(l)}$ for nodes in $\mathbf{m}^{(l)}$.

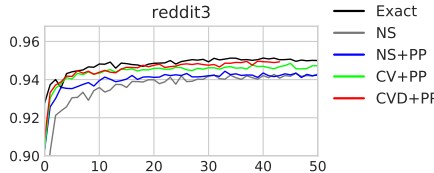

Figure 6: Comparison of validation accuracy with respect to number of epochs for 3-layer GCNs.

Table 5: Time to reach 0.95 testing accuracy.

| Alg. | Valid. acc. | Epochs | Time (s) | Sparse GFLOP | Dense TFLOP |
|---|---|---|---|---|---|
| Exact | 0.940 | **3.0** | 199 | 306 | 11.7 |
| NS | 0.940 | 24.0 | 148 | 33.6 | 9.79 |
| NS+PP | 0.940 | 12.0 | 68 | **2.53** | 4.89 |
| CV+PP | 0.940 | 5.0 | **32** | 8.06 | **2.04** |
| CVD+PP | 0.940 | 5.0 | 36 | 16.1 | 4.08 |

### E.2 TRAINING WITH THE CVD ESTIMATOR

Training with the CVD estimator is similar with the CV estimator, except it runs two versions of the network, with and without dropout, to compute the samples $H$ and their mean $\mu$ of the activation. The matrix $\bar{P}_{v,v'} = \hat{P}_{v,v'}/\sqrt{|\mathbf{n}(v,1)|}$, where $|\mathbf{n}(v,1)|$ is the degree of node $v$.

## F EXPERIMENT FOR 3-LAYER GCNS

We test 3-layer GCNs on the Reddit dataset. The settings are the same with 2-layer GCNs in Sec. 5.3. To ensure the exact algorithm can run in a reasonable amount of time, we subsample the graph so that the maximum degree is 10. The convergence result is shown as Fig. 6, which is similar with the two-layer models. The time consumption to reach 0.94 testing accuracy is shown in Table 5.

## G JUSTIFICATION OF THE INDEPENDENT GAUSSIAN ASSUMPTION

### G.1 RESULTS FOR 2-LAYER GCNS

We justify the independent Gaussian assumption in Sec. 4.3 by showing that for a 2-layer GCN with the first layer pre-processed, the neighbor's activations are independent. Without loss of generality, suppose that we want to compute $z_1^{(2)}$, and the neighbors of node 1 are $1, \ldots, D$. By Eq. (1), $h_v^{(1)} = \sigma\left((\phi_v \circ u_v^{(0)})W^{(0)}\right)$ is a random variable with respect to $\phi_v$, where $\phi_v \sim \text{Bernoulli}(p)$ is the dropout mask and $u_v^{(0)} = (PH^{(0)})_v$. The indepedent Gaussian assumption states that

1. $h_v^{(1)}$ is a Gaussian random variable with diagonal covariance;

2. $h_v^{(1)}$ and $h_{v'}^{(1)}$ are independent, for $v \neq v'$.

Assumption 1 is not GCN-specific and is discussed in Wang & Manning (2013), we now prove assumption 2 by the following lemma.

**Lemma 3.** *If $a$ and $b$ are independent random variables, then their transformations $f_1(a)$ and $f_2(b)$ are independent.*

Because for any event $A$ and $B$, $P(f_1(a) \in f_1(A), f_2(b) \in f_2(B)) = P(a \in A, b \in B) = P(a \in A)P(b \in B) = P(f_1(a) \in f_1(A))P(f_2(B) \in f_2(B))$, where $f_1(A) = \{f_1(a)|a \in A\}$ and $f_2(B) = \{f_2(b)|b \in B\}$.

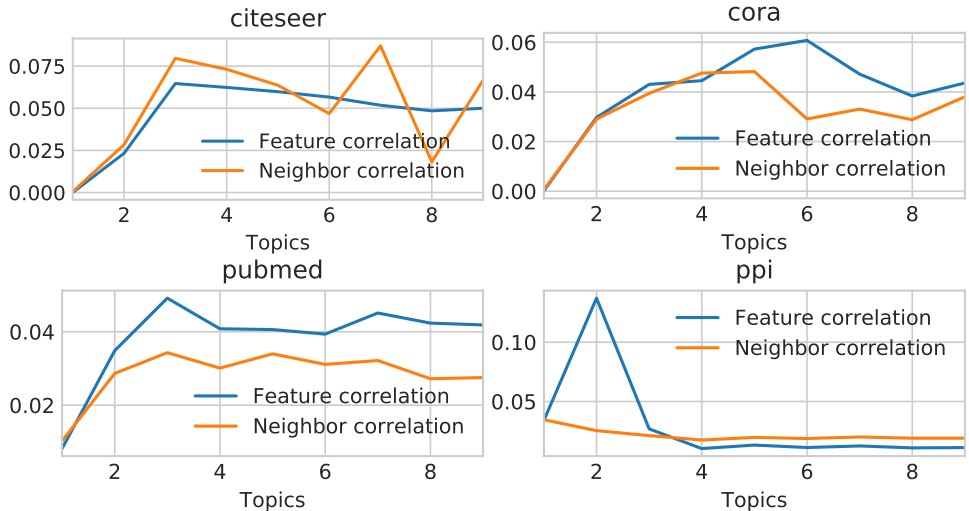

Figure 7: Average feature and neighbor correlations in a 10-layer GCN.

Let $h_v^{(1)} = f_1(\phi_v) := \sigma\left((\phi_v \circ u_v^{(0)})W^{(0)}\right)$ and $h_{v'}^{(1)} = f_1(\phi_{v'}) := \sigma\left((\phi_{v'} \circ u_{v'}^{(0)})W^{(0)}\right)$, because $\phi_v$ and $\phi_{v'}$ are independent Bernoulli random variables, $h_v^{(1)}$ and $h_{v'}^{(1)}$ are independent.

The result can be further generalized to deeper models. If the receptive fields of two nodes does not overlap, they should be independent.

### G.2 EMPIRICAL RESULTS FOR DEEPER GCNS

Because we only sample two neighbors per node, the sampled subgraph is very close to a graph with all its nodes isolated, which reduces to the MLP case that Wang & Manning (2013) discuss.

We empirically study the correlation between feature dimensions and neighbors. The definition of the correlation between feature dimensions is the same with Wang & Manning (2013). For each node $v$ on layer $l$, we compute the correlation between each feature dimension of $h_v^{(l)}$

$$\text{Cov}_{ij}^{(l,v)} := \mathbb{C}[h_{vi}^{(l)}, h_{vj}^{(l)}]$$

$$\text{Corr}_{ij}^{(l,v)} := \frac{\text{Cov}_{ij}^{(l,v)}}{\sqrt{\text{Cov}_{ii}^{(l,v)}}\sqrt{\text{Cov}_{jj}^{(l,v)}}},$$

where $i$ and $j$ are the indices for different hidden dimensions, and $\mathbb{C}[X, Y] = \mathbb{E}[(X - \mathbb{E}X)(Y - \mathbb{E}Y)]$ is the covariance between two random variables $X$ and $Y$. We approximate $\text{Cov}_{ij}^{(l,v)}$ with 1,000 samples of the activations $h_{vi}^{(l)}$ and $h_{vj}^{(l)}$, by running the forward propagation 1,000 times with different dropout masks. We define the *average feature correlation on layer $l$* to be $\text{Cov}_{ij}^{(l,v)}$ averaged by the nodes $v$ and dimension pairs $i \neq j$.

To compute the correlation between neighbors, we treat each feature dimension separately. For each layer $l + 1$, node $v$, and dimension $d$, we compute the correlation matrix of all the activations $\{h_{id}^{(l)} | i \in \bar{\mathbf{n}}^{(l)}(v)\}$ that are needed by $h_{vd}^{(l+1)}$, where $\bar{\mathbf{n}}^{(l)}(v) = \{i | \hat{P}_{vi}^{(l)} \neq 0\}$ is the set of subsampled neighbors for node $v$:

$$\text{Cov}_{ij}^{(l,v,d)} := \mathbb{C}[h_{id}^{(l)}, h_{jd}^{(l)}]$$

$$\text{Corr}_{ij}^{(l,v,d)} := \frac{\text{Cov}_{ij}^{(l,v,d)}}{\sqrt{\text{Cov}_{ii}^{(l,v,d)}}\sqrt{\text{Cov}_{jj}^{(l,v,d)}}},$$

where the indices $i, j \in \bar{\mathbf{n}}^{(l)}(v)$. Then, we compute the average correlation of all pairs of neighbors $i \neq j$.

$$\text{AvgCorr}^{(l,v,d)} := \frac{1}{\left|\bar{\mathbf{n}}^{(l)}(v)\right| \left(\left|\bar{\mathbf{n}}^{(l)}(v)\right| - 1\right)} \sum_{i \neq j} \text{Corr}_{ij}^{(l,v,d)},$$

and define the *average neighbor correlation on layer $l$* as $\text{AvgCorr}^{(l,v,d)}$ averaged over all the nodes $v$ and dimensions $d$.

We report the average feature correlation and the average neighbor correlation per layer, on the Citeseer, Cora, PubMed and PPI datasets. These quantities are too expensive to compute for NELL and Reddit. On each dataset, we train a GCN with 10 graph convoluation layers until early stopping criteria is met, and compute the average feature correlation and the average neighbor correlation for layer 1 to 9. We are not interested in the correlation on layer 10 because there are no more graph convolutional layers after it. The result is shown as Fig. 7. As analyzed in Sec. G.1, the average neighbor correlation is close to zero on the first layer, but it is not exactly zero due to the finite sample size for computing the empirical covariance. There is no strong tendency of increased correlation as the number of layers increases, after the third layer. The average neighbor correlation and the average feature correlation remain on the same order of magnitude, so bringing correlated neighbors does not make the activations much more correlated than the MLP case (Wang & Manning, 2013). Finally, both correlations are much smaller than one.

