# OpenReview forum: "Stochastic Training of Graph Convolutional Networks"
_ICLR.cc/2018/Conference — Reject_

### Official Review · AnonReviewer3 · 2017-11-19
**A new training method of graph convolutional networks. Good but there are some errors.**

**Rating:** 3
**Confidence:** 4

**Review:**

This paper proposes a new training method for graph convolutional networks. The experimental results look interesting. However, this paper has some issues.

This paper is hard to read. There are some undefined or multi-used notations. For instance, sigma is used for two different meanings: an activation function and variance. Some details that need to be explained are omitted. For example, what kind of dropout is used to obtain the table and figures in Section 5? Forward and backward propagation processes are not clearly explained

In section 4.2, it is not clear why we have to multiply sqrt{D}. Why should we make the variance from dropout sigma^2?

Proposition 1 is wrong. First, \|A\|_\infty should be max_{ij} |A_ij| not A_{ij}. Second, there is no order between \|AB\|_\infty and \|A\|_\infty \|B\|_\infty. When A=[1 1] and B is the transpose matrix of A, \|AB\|_\infty =2 and \|A\|_\infty \|B\|_\infty = 1. When, A’=[1 -1] and B is the same matrix defined just before, \|A’ B \|_\infty = 0 and \|A’\|_\infty \|B\|_\infty =1. So, both \|AB\|_\infty \le \|A\|_\infty \|B\|_\infty and \|AB\|_\infty \ge \|A\|_\infty \|B\|_\infty are not true. I cannot believe the proof of Theorem 2.

---

> ### Author Response · Authors · 2018-01-05
> **Added clarifications and fixed the typos. We believe our technical contribution is significant, please reconsider it given the clarifications!**
>
> Thanks for your review! The review mostly concerns about some unclarified details and typos in the paper. We addressed these concerns below. Meanwhile, the review does not mention any aspects about technical contribution itself. We think stochastic training for graph convolutional networks is very important for scaling up neural networks towards practical graphs and helping develop more expressive models. Our approach is significant both practically and theoretically. We can compute approximate gradients for GCNs at a cost similar with MLPs, while losing little convergence speed. We also provide new theoretical guarantees to reach the same training and testing performance of the exact algorithm. After we clarified all the mentioned details and typos, could you please also assess the work based on the technical contribution? We are happy to give more clarifications if needed.
>
> Q1: This paper is hard to read. There are some undefined or multi-used notations. For instance, sigma is used for two different meanings: an activation function and variance. Some details that need to be explained are omitted. For example, what kind of dropout is used to obtain the table and figures in Section 5? Forward and backward propagation processes are not clearly explained.
>
> We change the notation for variance from \sigma^2 to s^2. Throughout the paper we only have one kind of dropout (Srivastava et al., 2014), which randomly zeros out features. The dropout operation in our paper is already explained in Eq. (1). We add a pseudocode in appendix E to explain the forward and backward propagation processes. Basically, forward propagation is defined using Eq. (5) and Eq. (6) and backward propagation is simply computing the gradient of the objective with respect to the parameters automatically.
>
> Q2: In section 4.2, it is not clear why we have to multiply sqrt{D}. Why should we make the variance from dropout sigma^2?
>
> We multiply sqrt{D} so that the approximated term and the original term have the same mean and variance, based on the case study under the independent Gaussian assumption in Sec. 4.3 (See Q3 of Reviewer 2 for the justification of the assumption). Under the assumption, the activations h_1, …, h_D are approximated by independent Gaussian random variables h_v~N(\mu_v, s_v^2), and the randomness comes from randomly dropping out features  from the feature vector x while computing h_v = activation(PWx). We define s_v^2 to be the variance of the Gaussian random variable p_1h_1+…+p_Dh_D. We separate p_1h_1+…+p_Dh_D as
> p_1(h_1-\mu_1)+…+p_D(h_D-\mu_D)    (which has zero mean)
> and
> p_1\mu_1+…+p_D\mu_D               (which is deterministic).
> We approximate the first term as sqrt{D}(h_v’-\mu_v’), where v’is selected uniformly from {1, …, D}. Because sqrt{D}(h_v’-\mu_v’) and p_1(h_1-\mu_1)+…+p_D(h_D-\mu_D) have the same expected mean and variance, as shown in Appendix C.
>
> For short, without loss of generality we assume that \mu_1=…=\mu_D=0.
> Then, Var[h_1+…+h_D]=Var[h_1]+…+Var[h_D]=s_1^2+…+s_D^2 (because of independence). And
> E_{v’}[Var[sqrt{D} h_v’]]=E_{v’}[Ds_v’^2]= s_1^2+…+s_D^2.
>
>
> Q3: Proposition 1 is wrong. First, \|A\|_\infty should be max_{ij} |A_ij| not A_{ij}. Second, there is no order between \|AB\|_\infty and \|A\|_\infty \|B\|_\infty. I cannot believe the proof of Theorem 2.
>
> Thanks for pointing out. The correct version should be \|AB\|_\infty <= col(A) \|A\|_\infty \|B\|_\infty, where col(A) is the number of columns of A (or number of rows of B). We updated Proposition 1 and its proof. Note that the constant col(A) is absorbed and does not affect the proof of Theorem 2.
>
> Besides the proof, Theorem 2 is also verified empirically in Fig. 2, where the algorithm using CV’s approximated gradients (CV+PP) has almost an overlapping convergence curve with the algorithm using exact stochastic gradients (Exact).

---

### Official Review · AnonReviewer2 · 2017-11-26
**Interesting but not enough**

**Rating:** 4
**Confidence:** 4

**Review:**

The paper proposes a method to speed up the training of graph convolutional networks, which are quite slow for large graphs. The key insight is to improve the estimates of the average neighbor activations (via neighbor sampling) so that we can either sample less neighbors or have higher accuracy for the same number of sampled neighbors. The idea is quite simple: estimate the current average neighbor activations as a delta over the minibatch running average. I was hoping the method would also include importance sampling, but it doesn’t. The assumption that activations in a graph convolution are independent Gaussians is quite odd (and unproven).

Quality: Statistically, the paper seems sound. There are some odd assumptions (independent Gaussian activations in a graph convolution embedding?!?) but otherwise the proposed methodology is rather straightforward.

Clarity: It is well written and the reader is able to follow most of the details. I wish the authors had spent more time discussing the independent Gaussian assumption, rather than just arguing that a graph convolution (where units are not interacting through a simple grid like in a CNN) is equivalent to the setting of Wang and Manning (I don’t see the equivalence). Wang and Manning are looking at MLPs, not even CNNs, which clearly have more independent activations than a CNN or a graph convolution.

Significance: Not very significant. The problem of computing better averages for a specific problem (neighbor embedding average) seems a bit too narrow. The solution is straightforward, while some of the approximations make some odd simplifying assumptions (independent activations in a convolution, infinitesimal learning rates).

Theorem 2 is not too useful, unfortunately: Showing that the estimated gradient is asymptotically unbiased with learning rates approaching zero over Lipchitz functions does not seem like an useful statement. Learning rates will never be close enough to zero (specially for large batch sizes). And if the running activation average converges to the true value, the training is probably over. The method should show it helps when the values are oscillating in the early stages of the training, not when the training is done near the local optimum.

---

> ### Author Response · Authors · 2018-01-05
> **Remarks about importance sampling and justification for the independent Gaussian assumption**
>
> Thanks for the valuable comments. We address the detailed questions below.
>
> Q1: I was hoping the method would also include importance sampling:
>
> Importance sampling is a useful technique. There is another submission about importance sampling for graph convolutional networks [1]. We have some remarks regarding to importance sampling [1]:
> 1)	Our result is already close to the best we can possibly do, so importance sampling may only have marginal improvement. Despite using the much cheaper control-variate based gradient, we almost lost no convergence speed comparing with exact gradients (without neighbor sampling), according to Fig. 2 and Fig. 3.
> 2)	Importance sampling and our control variate & preprocessing are orthogonal techniques for reducing the bias and variance of the gradient.
> 3)	Our control variate based gradient estimator is asymptotically *unbiased*. As the learning rate goes to zero, our estimator yields unbiased stochastic gradient, *regardless of the neighbor sampling size*. On the other hand, the importance sampling based estimator is only *consistent*. It is unbiased *only when the neighbor sampling size goes to infinity*. This result is also shown experimentally: our work uses a very small neighbor sampling size (e.g., 2 neighbors for each node), while the neighbor sampling size of [1] is still hundreds. It takes only 50 seconds for our algorithm training on the largest Reddit dataset, while [1] takes 638.6 seconds.
> [1] FastGCN: Fast Learning with Graph Convolutional Networks via Importance Sampling. https://openreview.net/forum?id=rytstxWAW
>
> Q2: The assumption that activations in a graph convolution are independent Gaussians is quite odd (and unproven). I wish the authors had spent more time discussing the independent Gaussian assumption, rather than just arguing that a graph convolution (where units are not interacting through a simple grid like in a CNN) is equivalent to the setting of Wang and Manning (I don’t see the equivalence). Wang and Manning are looking at MLPs, not even CNNs, which clearly have more independent activations than a CNN or a graph convolution.
>
> The assumption makes some sense intuitively.
> 1)	If all the nodes are isolated, it reduces to the MLP case that Wang and Manning considered.
> 2)	In two-layer GCNs where the first layer is pre-processed, which is the most popular architecture, we can show that the neighbors’ activations are indeed independent with each other.
> 3)	In deeper GCNs, the correlations between neighbor’s may still be weak in our algorithm, because the sampled subgraph is very sparse (each node only picks itself and another random neighbor).
>
> Now we show that in two-layer GCNs where the first layer is pre-processed, the neighbors’ activations are indeed independent with each other (we added the discussions in Appendix G). Assume that we want to compute the gradient w.r.t. node “a” on the second layer, the computational graph looks like:
>
> Layer 2: a
> Layer 1: a b      (b is a random neighbor of a)
>
> By Eq. (3), h_a^1 = \sigma(Dropout(u^0_a) W^0) and h_b^1 = \sigma(Dropout(u^0_b) W^0), where U^0=PH^0. The independent Gaussian assumption states that h_a^1 and h_b^1 are independent. To show this, we need the Lemma (function of independent r.v. s): If a and b are independent r.v. s, then f_1(a) and f_2(b) are independent r.v. s
> https://math.stackexchange.com/questions/8742/are-functions-of-independent-variables-also-independent
>
> Let .* be the element-wise product. We have h_a^1 = f_1(\phi_a) := \sigma(\phi_a .* u^0_a) W^0 and h_b^1 := f_2(\phi_b) = \sigma(\phi_b .* u^0_b) W^0. Because the dropout masks \phi_a and \phi_b are independent, we know that h_a^1 and h_b^1 are independent by the lemma. The rest assumptions about the Gaussian approximation and the independence between feature dimensions are discussed in Wang and Manning.
>
> We admit that the independent Gaussian assumption is somewhat rough. However, we do not explicitly rely on the independent Gaussian assumption like Wang and Manning, where they directly compute the mean and variance for the activation, and manually derive update rules of the mean and variance after each layer. Our algorithm only requires the samples, and the algorithm itself can execute regardless of the distribution of activation.
>
> Overall, the assumption is more like a motivating case (in which the algorithm works perfectly) rather than a must-hold condition for the algorithm to work. In practice, our estimator does have smaller bias & variance than the estimator without control variates (Fig. 5), although the condition does not hold perfectly. Furthermore, our main theoretical result (Theorem 2) does not depend on the independent Gaussian assumption.

---

> > ### Author Response · Authors · 2018-01-05
> > **Significance, and non-asymptotic version of Theorem 2**
> >
> > Q3: The problem of computing better averages for a specific problem (neighbor embedding average) seems a bit too narrow.
> >
> > Graph convolutional networks (GCNs) are important extensions of CNNs to graph structured data. There are an increasing number of works applying GCNs to different graph-based problems including node classification, node embedding, link prediction and knowledge base completion, with state-of-the-art performance on a large proportion of these tasks. We believe that GCNs are revolutionizing graph-related areas just like CNNs did to the image-related tasks. Our method is general for different GCN variants across tasks, and thus is not narrow.
> >
> > Q4: The solution is straightforward:
> > Our solution is simple to implement and effective, which is an advantage to reproduce the results and build further extensions. But we believe the theory behind the simple updates is not straightforward. Unlike most variance reduction works, control variates bring stronger guarantees to the algorithm, besides just reducing the variance. Our algorithm is the first one that guarantees the testing accuracy and the convergence to local optimum, regardless of the neighbor sampling size. The simplicity, effectiveness and theoretical guarantee enable users easily adopt our technique to their models, and get good results.
> >
> > Q5: Theorem 2 is not too useful: Showing that the estimated gradient is asymptotically unbiased with learning rates approaching zero over Lipchitz functions does not seem like an useful statement. Learning rates will never be close enough to zero (specially for large batch sizes). And if the running activation average converges to the true value, the training is probably over. The method should show it helps when the values are oscillating in the early stages of the training, not when the training is done near the local optimum.
> >
> > We do have non-asymptotic version of Theorem 2 but we choose to present the asymptotic version for ease to understand in the submission. We can show that square norm of the gradient is proportional with 1/sqrt{N} with respect to the number of iterations \sqrt{N}, which is on the same order of the analysis by Ghadimi & Lan (2013) who used unbiased stochastic gradients (i.e., without sampling neighbors). Simple neighbor sampling or importance sampling does not have such a guarantee. The non-asymptotic result can be directly derived with the proof in appendix B. We replaced Theorem 2 with its non-asymptotic version in the revision.
> >
> > Our empirical results in Fig. 2 and Fig. 3 show that our method indeed helps in the early stages of the training. Despite using cheaper gradients by sampling neighbors, we almost have no loss of the convergence speed -- the number of epochs (or iterations) for our method (CV+PP & CVD+PP) to converge to a certain testing accuracy is almost the same for CV+PP/CVD+PP and Exact – which is the best we can possibly do.

---

### Official Review · AnonReviewer1 · 2017-11-26
**Existing training algorithms for graph convolutional nets are slow. This paper develops new novel methods, with a nice mix of theory, practicalities and experiments.**

**Rating:** 7
**Confidence:** 3

**Review:**

Existing training algorithms for graph convolutional nets are slow. This paper develops new novel methods, with a nice mix of theory, practicalities, and experiments.

Let me caution that I am not familiar with convolutional nets applied to graph data.

Clearly, the existing best algorithm - neighborhood sampling is slow as well as not theoretically sound. This paper proposes two key ideas - preprocessing and better sampling based on historical activations. The value of these ideas is demonstrated very well via theoretical and experimental analysis. I have skimmed through the theoretical analysis. They seem fine, but I haven't carefully gone through the details in the appendices.

All the nets considered in the experiments have two layers. The role of preprocessing to add efficiency is important here. It would be useful to know how much the training speed will suffer if we use three or more layers, say, via one more experiment on a couple of key datasets. This will help see the limitations of the ideas proposed in this paper.

In subsection 4.3 the authors prove reduced variance under certain assumptions. While I can see that this is done to make the analysis simple, how well does this analysis correlate with what is seen in practice? For example, how well does the analysis results given in Table 2 correlate with the standard deviation numbers of Figure 5 especially when comparing NS+PP and CV+PP?

---

> ### Author Response · Authors · 2018-01-05
> **Thanks for the review! Questions addressed**
>
> Thanks for the review! We addressed the comments below.
>
> Q1: How much the training speed will suffer if we use three or more layers, say, via one more experiment on a couple of key datasets.
>
> Thanks for the suggestion. We added the results for three-layer networks in appendix F on the Reddit dataset. The exact algorithm takes tens of thousands per epoch on the original graph (max degree is 128). We subsampled the graph so that the max degree is 10, CVD+PP is about 6 times faster than Exact to converge to 0.94 testing accuracy, and the convergences speed are reported in Fig. 6. The observations are pretty much the same, that control variate based algorithms are much better than those without control variates.
>
> Q2: Subsection 4.3: the authors prove reduced variance under certain assumptions. How well does this analysis correlate with what is seen in practice? For example, how well does the analysis results given in Table 2 correlate with the standard deviation numbers of Fig. 5 especially when comparing NS+PP and CV+PP?
>
> For models without dropout, the main theoretical result is Theorem 1, which states that CV+PP has zero bias & variance as the learning rate goes to zero, and the independent Gaussian assumption is not needed. Fig. 5 (top row) shows that the bias and variance of CV+PP are quite close to zero in practice, which matches the theoretical result.
>
> For models with dropout, we found that the standard deviations (Fig. 5 bottom right) of CV+PP and CVD+PP were greatly reduced from NS+PP, mostly because of the reduction of VMCA. The bias was not always reduced, which calls better treatment of the term (h_v - \mu_v) in Sec. 4.2. We do not use historical values for this term. Incorporating historical values for this term may further reduce the bias and generalize Theorem 2 (which does not rely on the independent Gaussian assumption) to the dropout case. This is one possible future direction.
>
> Q3: The paper may not appeal to a general audience since the ideas are very specific to graph convolutions, which itself is restricted only to data connected by a graph structure.
>
> Graph-structured data is prevalent, e.g., user-graphs, citation graphs, web pages, knowledge graphs, etc. Moreover, graphs are generalization of many data structures, e.g., an image can be represented by 2d lattices; and a document categorization task can be improved by utilizing the citations between them. We therefore think extending deep learning to graph-structured data is important.

---

### Public Comment · (anonymous) · 2017-11-27
**Very interesting paper and experimental results. The theoretical analysis needs a bit more work.**

The paper is addressing a very interesting problem with imminent importance and industrial impact.

In the proof of the unbiased estimator for gradient: that is the proof of theorem 1, two lines above equation (10), Z^(l) inside \sigma'(Z^(l)) also depends on the random sampling, no?

The situation is similar to doubly stochastic gradient descent:
Dai et al. Scalable Kernel Methods via Doubly Stochastic Gradients, NIPS 2016
https://arxiv.org/pdf/1407.5599.pdf
Line 5 Algorithm 1.
The analysis of the paper is able to take this source of bias into account.

The proof of the current paper could also be fixed accordingly.

---

> ### Author Response · Authors · 2018-01-05
> **Z^{(l)}_{CV} is deterministic**
>
> Thanks for your interest to our paper!
>
> After LI iterations, part 1 of Theorem 1 shows that the activations Z^{(l)}_{CV} and H^{(l)}_{CV} are DETERMINISTIC, because the history is already complete
>
> Z^{(l)}_{CV}=Z^{(l)}
> H^{(l)}_{CV}=H^{(l)}
>
> In other words, the forward propagation is deterministic and does not depends on \bar P^{(l)}. The only random thing is the gradient, because the backward propagation is stochastic.
>
> Therefore, sigma'(Z^{(l)}_{CV})=sigma'(Z^{(l)}) is deterministic. It does not depends on random sampling (after LI iterations).

---

### Author Response · Authors · 2018-01-06
**Summary of the revision**

We appreciate the valuable feedback from the reviewers. Based on your comments we made a few revisions.

1. We add experiments for >2 layers in appendix F as suggested by Reviewer 1.
2. We add some justifications of the independent Gaussian assumption in appendix G as suggested by Reviewer 2.
3. We replace Theorem 2 with its non-asymptotic version as suggested by Reviewer 2.
4. We fixed some typos as well as Proposition 1 as suggested by Reviewer 3.
5. We add pseudo-code for the CV and CVD algorithm in appendix E as suggested by Reviewer3.

Please see our response for individual comments for your questions. We are happy to provide more clarifications if needed.

---

### Public Comment · (anonymous) · 2018-01-21
**Speedup with vs. without preprocessing?**

I think this paper introduces interesting ideas to speed up the training of graph neural networks (and graph convolutional nets specifically) which could potentially have direct industrial impact.

I haven't yet fully worked through the mathematical motivation of the paper, but I was wondering how much impact the preprocessing of the first layer (in the paper denoted by "+PP") had on the timing results in Table 4? How much of the speedup of CV+PP and CVD+PP is due to the preprocessing (+PP) and how much is due to the actual variance reduction technique? I think this would be a very important distinction to make, as I think that the proposed preprocessing is not the major contribution of this paper (this seems more like an implementation detail).

---

> ### Author Response · Authors · 2018-01-22
> **Re: Speedup with vs. without preprocessing?**
>
> Thanks for you interest in our paper!
>
> For CV (without +PP), the running time is 95.85 seconds, while CV+PP takes 56 seconds in Table 4. Our implementation does not support CVD without PP. The improvement of PP is not as large as CV, but is reasonable given its simplicity. Furthermore, some theoretical results of CV and CVD depend on PP. For example, in Appendix G.1, we justify the independent Gaussian assumption in Sec. 4.3 by showing that node activations are independent in a two-layer GCN with PP.
>
> The original GCN model by Kipf and Welling does not support pre-processing. To enable pre-processing, we modify the GCN model by changing the order of dropout and multiplying the propagation matrix. We justify the modification and also show that the modification of the model does not affect the predictive performance in Table 3. Pre-processing reduces the number of graph convolution layers by one. Since the receptive field of GCN grows exponentially w.r.t. the number of layers, the removal of each layer has important impact to the time complexity. Such study should not be summarized as "implementation detail".

---

### Decision · Program_Chairs · 2018-01-29
**ICLR 2018 Conference Acceptance Decision**

**Decision:**

Reject

**Comment:**

The paper studies subsampling techniques necessary to handle large graphs with graph convolutional networks.  The paper introduces two ideas: (1) preprocessing for GCNs (basically replacing dropout followed by linear transformation with linear transformation followed by drop out); (2) adding control variates based on historical activations.  Both ideas seem useful (but (1) is more empirically useful than (2), Figure 4*). The paper contains a fair bit of math (analysis / justification of the method).

Overall, the ideas are interesting and can be useful in practice. However, not all reviewers are convinced that the methods constitute a significant contribution.  There is also a question whether the math has much value (strong assumptions - also, from interpretation, may be too specific to the formulation of Kipf & Welling making it a bit narrow?).  Though I share these feelings and recommend rejection, I think that the reviewers 2 and 3 were a bit too harsh, and the scores do not reflect the quality of the paper.

*Potential typo: Figure 4 -- should it be CV +PP rather than CV?

+ an important problem
+ can be useful in practical applications
+ generally solid and sufficiently well written
- significance not sufficient
- math seems not terribly useful